

# Effects of fire and grazing on biogeochemical cycles in Brazilian pastures using LPJmL5-Pasture-Burning

Marie Brunel[1,2], Stephen Wirth[1,3], Markus Drüke[1,4], Kirsten Thonicke[1], Henrique Barbosa[5], Jens Heinke[1], and Susanne Rolinski[1]

[1]Potsdam Institute for Climate Impact Research (PIK), Member of the Leibniz Association, P.O. Box 601203, 14412 Potsdam, Germany
[2]Department of Life Sciences, Humboldt University (HU), Invalidenstraße 42, 10115 Berlin, Germany
[3]Institute of Crop Science and Plant Breeding, Grass and Forage Science/Organic Agriculture, Kiel University Hermann-Rodewald-Str. 9, 24118, Kiel, Germany
[4]Deutscher Wetterdienst, Hydrometeorologie, Frankfurter Str. 135, 63067 Offenbach, Germany
[5]Department of Physics, University of Maryland, Baltimore County, Baltimore, MD, USA

**Correspondence:** Marie Brunel (brunelmarie1@gmail.com)

**Abstract.** Farmers across the world frequently use fire during the winter or dry season, to remove accumulated dead pasture biomass. These fire-management practices have profound effects on vegetation, soil nutrients, and biogeochemical cycles, yet they are rarely represented in process-based fire models embedded within Dynamic Global Vegetation Models (DGVMs). We couple the Chalumeau algorithm, which estimates expected burning dates, with the SPITFIRE module in the DGVM LPJmL

and enable the modelling of fire as a grassland management method. Using this model development, we examine the short- and long-term impacts of varying burning strategies, frequencies, and livestock densities across distinct regions, using Brazil as a case study. Our results show that integrating grazing and fire management leads to a gradual decline in vegetation carbon, accompanied by a substantial reduction of the ecosystem and soil nitrogen. This study emphasises the importance of incorporating such practices into DGVMs to enhance the accuracy of impact assessments for pasture management. Furthermore,

our findings call for improved data collection describing fire usage methods by farmers, as well as long-term measurements, particularly on vegetation, soil carbon and nitrogen development under burning practices.

## 1 Introduction

In seasonally dry biomes, it is customary for farmers to utilise fire to manage their land in the winter or dry season, which is commonly known as the dormant season and typically occurs in Brazil between May and November. These practices, based

on farmers' observations and assessment of field conditions (Mistry, 1998; Sorrensen, 2000; van der Werf et al., 2008), serve essentially for clearing the accumulated dead grassland biomass (Pillar and de Quadros, 1997; Mistry, 1998; Csiszar et al., 2012; Barlow et al., 2020). During the dormant season, when above-ground biomass usually dies off, there is a build-up of material that is burned by farmers. This practice is reported to promote the growth of herbaceous species with high nutritional value (Mistry, 1998; van der Werf et al., 2008). Fires additionally help to remove undesirable vegetation from these areas such

as shrubs and trees (Pivello, 2011; López-Mársico et al., 2019). From an economic perspective, fires are viewed as the most




affordable way of achieving these purposes with the least possible human labour and investment costs (Mistry, 1998; Pivello et al., 2021). Nonetheless, notable disadvantages exist. The practice of fire impacts to the deterioration of the atmosphere through the emission of greenhouse gases, smoke, and particulate matter, which may have adverse effects on the health of the local population and impact local and global climate (Freitas et al., 2005; Ignotti et al., 2010; Nawaz and Henze, 2020).

Additionally, such practices increase the risk of wildfires especially in the Amazon region (Cano-Crespo et al., 2015; Brando et al., 2020). These out-of-control human-started fires often spill over into other vegetative layers, most frequently, the driest edges of residual woodland patches, which are highly susceptible to burn (Achard et al., 2002; Nepstad et al., 2008; Bonaudo et al., 2014; Barlow et al., 2020). The situation is even more complicated through land clearing and fragmentation, which heighten the perimeter of contact between cultivated land and natural vegetation so that the potential danger of fire outbreaks is

increased (Cochrane and Laurance, 2002; Cochrane, 2009). With all these factors, achieving effective outcomes from fire use requires careful timing and meticulous planning.

Variations of burning methods can be observed with respect to when, where, and how often the fires are set. The decision of farmers to set fire is determined primarily by the state of the vegetation cover, and hence by climate and its seasonal variation (van der Werf et al., 2008; Brunel et al., 2021). In Brazil, the climate can be divided into two distinct regions: areas like

the Pampas and the south of the Atlantic Forest are influenced by temperature seasonality, characterised by colder winters and hot summers, while the rest of the territory experiences precipitation seasonality, with noticeable wet and dry seasons. Farmers typically burn during the dormant season – either the winter months in temperature-seasonal regions or the dry season in precipitation-seasonal areas – when vegetation growth slows, and dead biomass accumulates. Fire may be set just before anticipated rain to help control fire spread, though this can also contribute to increased soil erosion. It is applied very often,

e.g. every two years in the Cerrado, as well as less frequent, e.g. every three to ten years in the Amazon (Pivello et al., 2021).

It is estimated that 40% of the annual burned area in South America can be attributed to fire practices on pastures (Rabin et al., 2015) excluding escaping fire dynamics. In Brazil, pastures account for around 20% of the total burned area. The Cerrado region has experienced an increase in grassland burning, which now represents approximately 35% of the total regional burned area, despite the fact that pastureland area was roughly constant in the past 20 years (MapBiomas, 2021). Recently, there has

been a growing interest within the scientific community in grassland ecosystems and the role of fire practices, highlighting the necessity to better understand their specific issues (Overbeck et al., 2015, 2024). Santos et al. (2023) argue that management of pastures is important not only for carbon sequestration but also for soil fertility. Studies show that in order to ensure the sustainability of pastures in the Brazilian savannahs, there is a need to monitor the carbon stocks (Fronza et al., 2024). For example, estimated carbon stocks provide important benchmarks for evaluating the impacts of different management practices.

In the Cerrado, the average carbon stocks in the soil (0–20 cm) and aboveground biomass are estimated at approximately 31 MgC ha$^{-1}$ and 4 MgC ha$^{-1}$, respectively, based on modelling studies with the CENTURY model (Santos et al., 2023). Monitoring these stocks over time would allow for assessing whether management practices, such as grazing or fertilisation, are helping to maintain or improve these levels.

In temperate pasture areas in North America, fire affects soil properties, especially nitrogen dynamics (Neary and Leonard,

2020). For example, the annual burning of tallgrass prairies in the Great Plains of the central United States has led to a notable



decrease in soil organic nitrogen and microbial biomass along with higher carbon to nitrogen (C:N) ratios in soil organic matter (Ojima et al., 1994). Although burning initially boosts soil fertility by elevating nutrient amounts and improving factors such as pH, exchangeable cations, and $NO_3$-N, these benefits often fade over time, ultimately returning to or falling below pre-burn levels (Mapiye et al., 2008).

Understanding fire-vegetation interactions is critical for predicting carbon and nitrogen fluxes, land management impacts, and vegetation dynamics. Modelling approaches using Dynamic Global Vegetation Models (DGVMs) are helpful in ascertaining the fire practices' benefits and drawbacks, offering insights into their ecological implications. However, DGVMs struggle to accurately capture human-caused ignitions in natural vegetation and many neglect fires on managed land. This remains difficult because the onset of the burning season in pastures depends upon the choice of farmers, considering vegetation conditions
and current weather. They usually decide on an appropriate burning period mostly based on their experience and the purpose of carrying out the burning, taking into account climatic but also social and economic factors (Mistry, 1998; van der Werf et al., 2008). For example, the Fire Including Natural & Agricultural Lands (FINAL) model incorporates cropland and pasture burning from natural fires through a dedicated module (Rabin et al., 2018). It considers fire seasonality, fire occurrence rates, and land cover data to simulate burned areas. However, the climatological approach of the model relies on only nine years of
observational data, which inevitably limits its ability to capture interannual variability. While this limitation is understandable given the constraints of available data, it does pose the question on the performance of the calibrated parametrisations under long-term historical simulations or future scenario projections.

To overcome these limitations, our research aims to go a step further and include the decision processes of the farmers into the algorithm. This approach seeks to improve the representation of region-specific fire ignitions and their interaction with
pasture biogeochemistry, providing a more nuanced understanding of fire dynamics on managed lands. The DGVM Lund-Potsdam-Jena managed Land (LPJmL) (Bondeau et al., 2007; Schaphoff et al., 2018a, b), simulates natural vegetation as well as managed land, including pastures, with integrated carbon and nitrogen cycles (von Bloh et al., 2018a, b). The model features the SPITFIRE module (Thonicke et al., 2010), which simulates both natural and human-caused wildfires in natural vegetation in the absence of firefighting or other fire management techniques. While SPITFIRE is calibrated to better capture the spatial
and temporal patterns of fire in South American biomes (Drüke et al., 2019), it does not explicitly account for region-specific fire management practices, such as pasture burning in biomes like the Cerrado or the Pampas. Fire ignition is driven by lightning and population density, which does not reflect the ignition dynamic of fire practices on grasslands.

To better assess fire regimes also in the agricultural context, we developed the Chalumeau algorithm to estimate expected burning dates based on management strategies and precipitation or temperature data (Brunel et al., 2021). In this study, we
coupled Chalumeau as the fire ignition mechanism with the SPITFIRE module, adjusted specifically to grassland, in LPJmL to simulate fire practices on pastures and quantify its feedback with soil carbon and nitrogen fluxes. We prescribe burned area and implement management strategies such as specific burning frequencies, e.g. every 2, 5, or 10 years, which will allow us to investigate the impacts of different management practices. This coupling attempts to improve the accuracy of modelling fire practices on pastures by better representing annual seasonality and interannual variability of burning dates, which remains to



be thoroughly tested. Hence, giving an opportunity to evaluate their consequences over a wider range of spatial and temporal scales.

The aim of this study is to analyse the short- and long-term impacts of fire practices coupled with grazing activity on pasture scale, focusing on dimensions vegetation status and productivity, field productivity, soil nutrient levels, and nitrogen emissions. We assess the field productivity by examining the vegetation development and the dry matter intake as it represents the yield. By

analysing the C:N ratio in both leaves and soil pools, we can identify fertilisation effects due to potential nitrogen enrichment. Additionally, studying the ecosystem and soil nitrogen cycle enhances our understanding of how these effects interconnect and their underlying dynamics. Through this comprehensive analysis, we provide insights into how pasture burnings influence grassland ecosystems across Brazilian regions, supporting the urge for better understanding and consideration of fire practices on pastures and their impacts.

## 2    Methods


The methods section provides an overview of the LPJmL modelling framework, introduces the SPITFIRE grassland module used for simulating fire dynamics, and details the model configuration, the experimental setup and the post-processing employed in this study.

### 2.1    LPJmL modelling concept

#### 2.1.1    Overview


The LPJmL model simulates the carbon, nitrogen, and water cycles as well as vegetation dynamics depending on climatic conditions, soil characteristics, and management methods. The photosynthesis is represented by a simplified Farquhar approach, as typical for global models (Collatz et al., 1991, 1992; Farquhar et al., 1980). Resulting gross primary production (GPP) and the auto- and heterotrophic respiration constitute the carbon fluxes into and out of the vegetation-soil continuum and impact

the different carbon reservoirs composed of: leaves, sapwood, heartwood, roots, storage organs, litter, and soil. Additionally, other processes also contribute to these fluxes: fire emissions and harvesting or grazing act as losses, removing carbon from the system, while returned manure from grazing animals contributes as an influx, adding carbon back into the soil pool. The main processes of the water balance – precipitation, interception, percolation, evaporation, transpiration, and run-off – are captured following Schaphoff et al. (2018a).

The model is usually applied at a resolution of $0.5° \times 0.5°$ latitude and longitude. Every grid cell is split into spatial units, so-called stands, which possess separate specific carbon, nitrogen, and water budgets. The soil is characterised by a depth of 3 m divided into 5 layers with respective thicknesses of 0.2, 0.3, 0.5, 1 and 1 m.





### 2.1.2 Managed grassland and grazing

In the LPJmL model, there are 12 crop functional types (CFTs) (Bondeau et al., 2007; Müller and Robertson, 2013) which

can be cultivated under rainfed or irrigated conditions (Rost et al., 2008; Jägermeyr et al., 2015) on specifically assigned stands. For this study, we focused exclusively on rainfed and managed grasslands. In LPJmL, they are established through the inclusion of three herbaceous plant functional types (PFTs). Plant growth, vegetation, water, carbon, and nitrogen dynamics are calculated for one representative average individual for every PFT. PFTs compete for light, available soil water, mineral nitrogen, and space. Carbon assimilation via photosynthesis, biological nitrogen fixation (BNF), plants' nitrogen uptake, and

water consumption are parametrised at the leaf level. Values are determined at daily time steps, similar to plant and soil respiration. Harvest events are modelled as the removal of leaves by mowing or grazing. Grass biomass is calculated on a daily basis, following the allocation of absorbed carbon as described by Rolinski et al. (2018).

For simulating continuous grazing, a fixed amount of leaf carbon is consumed every day per livestock unit (LSU), equivalent to one bovine of 650 kg body weight. The stocking density is set for each grid cell. To prevent long-term damage to the pasture,

grazing is restricted to times when there is at least 5 gC m$^{-2}$ of leaf carbon available, following the assumption that livestock are removed or fed externally at periods of low biomass. The daily grazing requirement is given at 4 kgC per LSU per day. 25% of the grass ingested is returned to the field in the form of manure and incorporated into the fast soil carbon pool (Soussana et al., 2014).

### 2.1.3 Soil nitrogen pools

In the LPJmL model, the nitrogen soil organic matter (SOM-N) is represented by the soil nitrogen pool, while the combined nitrate ($NO_3^-$) and ammonium ($NH_4^+$) soil pools encompass the nitrogen soil mineral matter (SMM-N). Each of these pools is calculated for individual soil layers and aggregated across the soil column for assessment in this study. The primary nitrogen inputs to the SOM-N pool originate from plant litterfall and manure. The SMM-N pool directly receives nitrogen from fire via ash deposition into the $NO_3^-$ pool, manure and BNF into the $NH_4^+$ pool, and atmospheric deposition into the $NO_3^-$ and $NH_4^+$

pools. Vegetation then assimilates nitrogen from this reservoir through uptake and nitrogen allocation processes.

These two nitrogen pools are interconnected through the dynamics of immobilisation and mineralisation. Immobilisation involves the conversion of inorganic nitrogen into organic forms by soil microorganisms. This process transforms SMM-N to SOM-N, making it unavailable to plants. Conversely, mineralisation is the microbial decomposition of organic nitrogen into inorganic forms, releasing nitrogen that plants can readily absorb.

## 2.2 SPITFIRE grassland module

SPITFIRE (SPread and InTensity of FIRE, Thonicke et al., 2010) is a process-based fire module that is used in LPJmL to represent fire disturbances. It models fire dynamics by simulating the different stages of fire: ignition, fire danger, spread, and its impacts on the ecosystem. Human activities and lightning as potential sources of combustion are both taken into account. Fire danger is estimated by the Nesterov index (Nesterov, 1949), which is determined with daily maximum and





dew point temperatures along its scaling factors for specific PFTs. This feature was improved by Drüke et al. (2019), who incorporated the water vapour pressure deficit (VPD) into the estimation of fire danger, with a particular focus on the Caatinga and the Cerrado biomes in Brazil. The forward rate of spread is calculated employing Rothermel's equations. The module then combines fire ignitions, danger, and spread to provide the extent of the area burned, fire-related carbon emissions, and plant mortality. Notably, it only simulates wildfires in natural vegetation and could be applied to fires in managed zones like

agricultural and pasture lands.

The following subsections outline the modifications necessary to utilise SPITFIRE for simulating fires on pasture.

### 2.2.1 Burning date

Contrary to wildfires in natural vegetation, managed grassland fires are intentionally planned in advance and are ignited by farmers at some predetermined time. The annual burning date is estimated through the 'Chalumeau' rule-based algorithm

(Brunel et al., 2021), which takes into account the climatic conditions and the burning strategy. Although the seasonal conditions restrict potential time windows for burning, the modelling scheme (Fig. 1) has to incorporate assumptions on human judgement processes. 'Chalumeau' calculates first the dormant season ($DS$) for every grid cell. The determination of winter or dry season is based on daily temperature or daily precipitation depending on the seasonality type of each location (Waha et al., 2012). The burning date is extracted from the duration of the $DS$ and a predefined burning strategy. Four burning strategies are

implemented to describe the setting of fire before or after the end of the $DS$ in order to cover the wide range of choices across Brazil (Brunel et al., 2021): 'early season', 'late season', 'end season' and 'early spring'.

### 2.2.2 Fuel condition

This subsection details the estimation of fuel conditions based on litter moisture. Since fuel within SPITFIRE Grassland is herbaceous, adjustments are incorporated to better represent the expected fire behaviour specific to this vegetation type.

**Burned area**

The burned area is computed as an output within the SPITFIRE module along with fire characteristics (Thonicke et al., 2010). However, for fire practices, we assume that farmers set the area to be burned as an objective target for the management of the pasture. Thus, in SPITFIRE grassland the surface burned is given as a parameter, expressed as a fraction of the total area of the stand.

**Daily litter moisture ratio and moisture extinction**

The litter moisture ratio $\omega_n$ describes the moisture status of the surface litter within the interval $[0, 1]$. $\omega_n$ is calculated as the ratio between the litter humidity and the litter's water-holding capacity of the surface (Lutz et al., 2019). A low value for the ratio indicates a completely dry litter and a high value means a water-saturated litter.

The moisture extinction represents the inverse of the fractional humidity content of a fuel pool that prevents fire from

developing. It is 0 at full fuel humidity and increases to 1 for entirely dry fuel. In the case of fire practices, however, we assume fires are initiated by farmers. Hence, the ignition is not dependent on fuel humidity since extra energy and time are added until burning objectives are executed. Therefore, the moisture extinction value for dead fuel and live grass is set to 1.



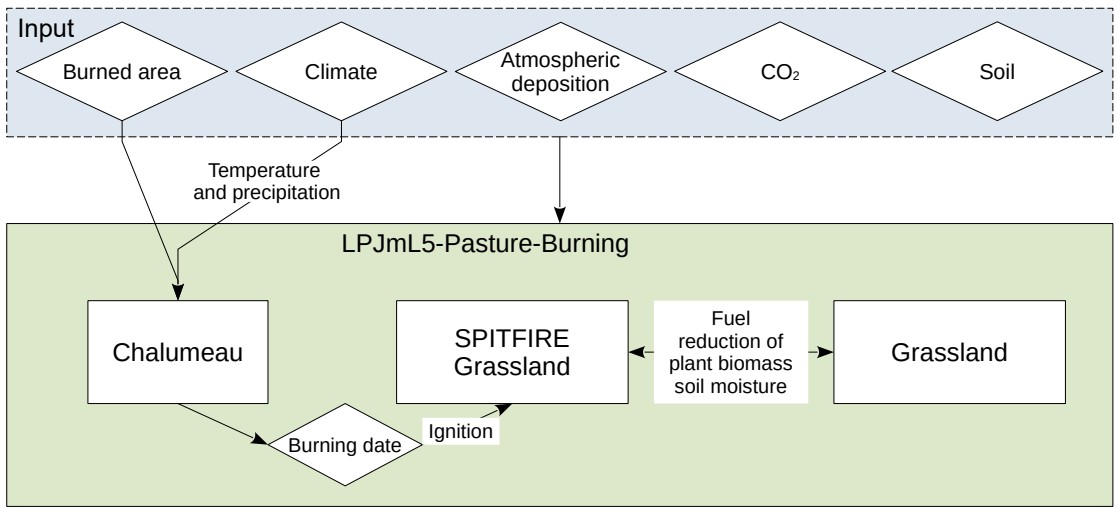

**Figure 1.** LPJmL5-Pasture-Burning model's conceptual scheme depicting the interaction of environmental inputs (soil, atmospheric nitrogen deposition, $CO_2$, burned area and climate) in relation to grassland processes and fire modules Chalumeau and SPITFIRE Grassland.

**Dead and live fuel consumption**

SPITFIRE categorises fuel into four distinct types: 1-hour, 10-hours, 100-hours, and 1000-hours fuel classes, which indicate
the fuel's relative burning potential in different vegetation types. The 1-hour fuel class includes materials that ignite and burn
quickly, such as living herbaceous biomass or leaf and small woody litter components.

Only the 1-hour fuel class of the four implemented in SPITFIRE is considered in the SPITFIRE grassland version. The
computation of the dead and live fuel consumption is based on the methodology of Peterson and Ryan (1986) and previous
work on SPITFIRE module for natural vegetation (Thonicke et al., 2010). Specifically, the functions dead-fuel-consumption,
fuel-load, and fuel-consumption are employed in this version to model grassland fire dynamics. The amount of fuel utilised $F_C$
in gC m$^{-2}$, is calculated depending on the fuel moisture $F_m$, fuel load $F_l$ in gC m$^{-2}$, and the fire fraction $Fire_{frac}$ (Eq. 1).

$$F_C = \left\{ \begin{array}{ll} 1.0, & \text{if} \quad F_m \leq 0.32 \\ 1.2 - 0.62 \cdot F_m, & \text{if} \quad 0.32 < F_m \leq 0.68 \\ 2.45 - 2.45 \cdot F_m, & \text{if} \quad F_m > 0.68 \end{array} \right\} \cdot F_l \cdot Fire_{frac} \tag{1}$$

For the dead fuel consumption calculation, equation 1 is employed, taking the daily litter moisture ratio $\omega_n$ as fuel moisture
$F_m$ indication. The live fuel computation accounts for the moisture content in living vegetation, which is influenced by the soil



moisture available in the topsoil layer, as described by Thonicke et al. (2010). The fuel load is composed of the carbon and nitrogen content of leaves. The fire fraction is determined by the burned area.

## 2.3 Model setup and input parameters

### 2.3.1 Input data sources

The NASA Global Land Data Assimilation System (GLDAS, Rodell et al., 2004; NASA, 2015) provides daily average temperature, radiation and total daily precipitation data from 1948 to 2019. These datasets are initially made available with a temporal resolution of three hours and a spatial resolution of $0.25° \times 0.25°$ for latitude and longitude. For our analysis, we aggregated the data to a daily temporal resolution and a spatial resolution of $0.5° \times 0.5°$ using the Climate Data Operator software (CDO, Schulzweida, 2019), applying a weighted average approach with the size of each cell as the weight. Characteristics of soils

are sourced from the Harmonised World Soil Database (version 1.2) (Fischer et al., 2012). The model incorporates historical atmospheric nitrogen deposition data (Tian et al., 2018) and global annual atmospheric $CO_2$ concentrations levels derived from the Mauna Loa station (Le Quéré et al., 2015).

### 2.3.2 Model configuration and experimental setup

Model experiments using LPJmL are performed at selected locations across Brazil with at least one site per biogeographic
region to capture the diverse climate, vegetation and soil conditions of the country (Fig. 2). These specific study sites are chosen to represent the diversity of conditions in Brazil, as applying the protocol to a single grid cell representing averaged conditions allows for a clearer focus on understanding the interactions between the system and the introduction of fire practices. The designated study sites are selected to represent the average regional climate, based on GLDAS data (Sec. *2.3.1*), ensuring that their long-term annual averages for temperature, precipitation, and radiation fall within the mean $\pm$ one standard deviation.
With this process, one representative grid cell per region could be identified except for the Amazon, for which two locations are selected due to its heterogeneous climate.

The model simulations begin with a spin-up of 7000 years during which only natural vegetation is simulated to allow the carbon and nitrogen pools to reach equilibrium. Following this, pasture is introduced, and a subsequent spin-up of 390 years is conducted to account for the transition from natural vegetation to pastures. For both spin-up phases, the first 30 years (1948 to
1978) of the climate data and atmospheric deposition input data are utilised in cycles.

Burning practices form an important disturbance for the system. Therefore one additional pasture spin-up of 390 years is added with livestock and fire practices to simulate a pre-established disturbance scenario. The main simulation is then carried out over 70 years for both recent and pre-established scenario beginning in 1948 (Fig. 3).

The pre-established disturbance experiments (spin-up and core simulation phase) are conducted under a pasture management
scenario defined by various factors. These included grazing and livestock density, set at 0, 0.1, and 0.5 LSU ha$^{-1}$. These levels are determined through preliminary sensitivity analyses aimed at identifying a range that effectively captures the dynamics of fire management in grazed pastures. The primary goal is to examine how fire management interacts with grazing rather than



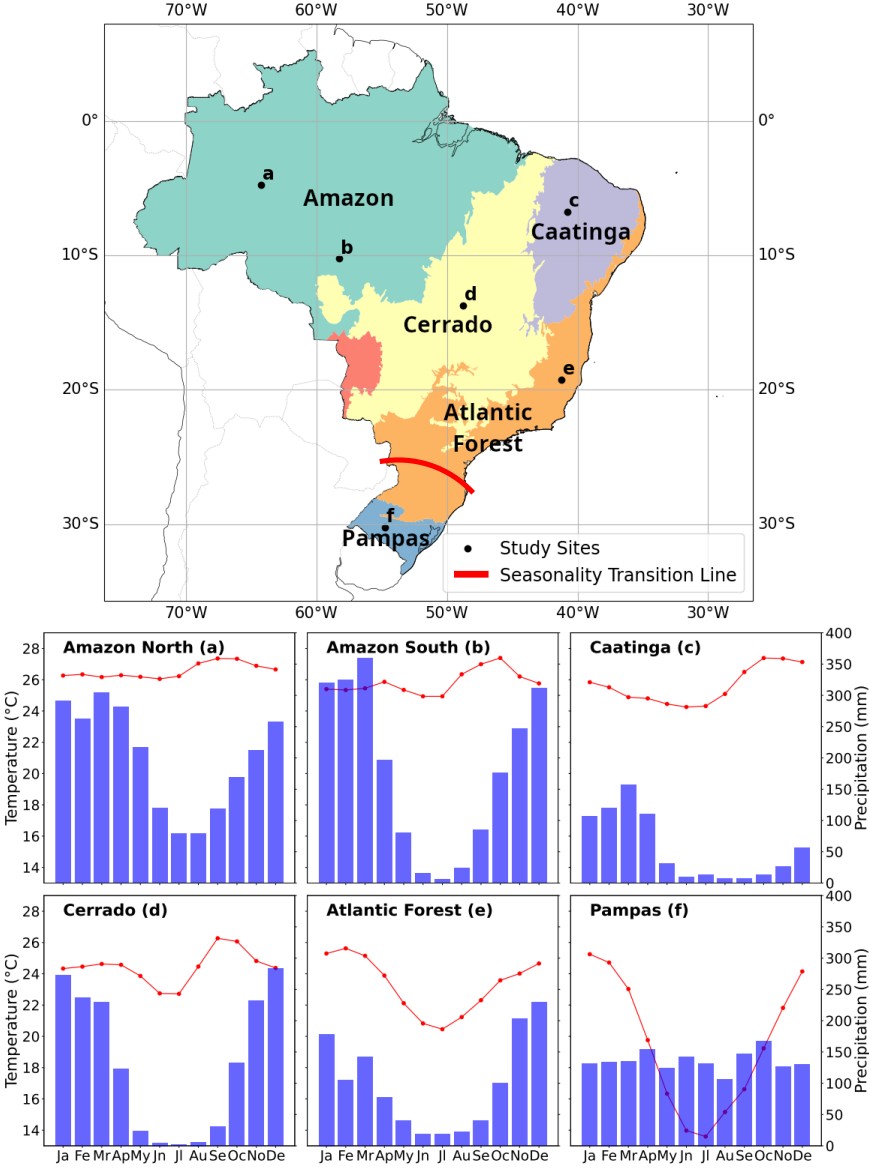

**Figure 2.** Map of Brazil illustrating the selected study sites within the major biomes (Amazon, Cerrado, Caatinga, Atlantic Forest, and Pampas) (Instituto Brasileiro de Geografia e Estatística, 2025). The red line delineates regions where seasonal variations are dominated by temperature (south) or precipitation (north). The lower panel shows the average monthly temperature (red line) and cumulative monthly precipitation (blue bars) for each region.

to achieve exact pasture yield estimates, which would require more detailed input on land-use and specific field management practices and is beyond the scope of this study. Fire practices, especially the frequency of burning, varied from every 0, 2,



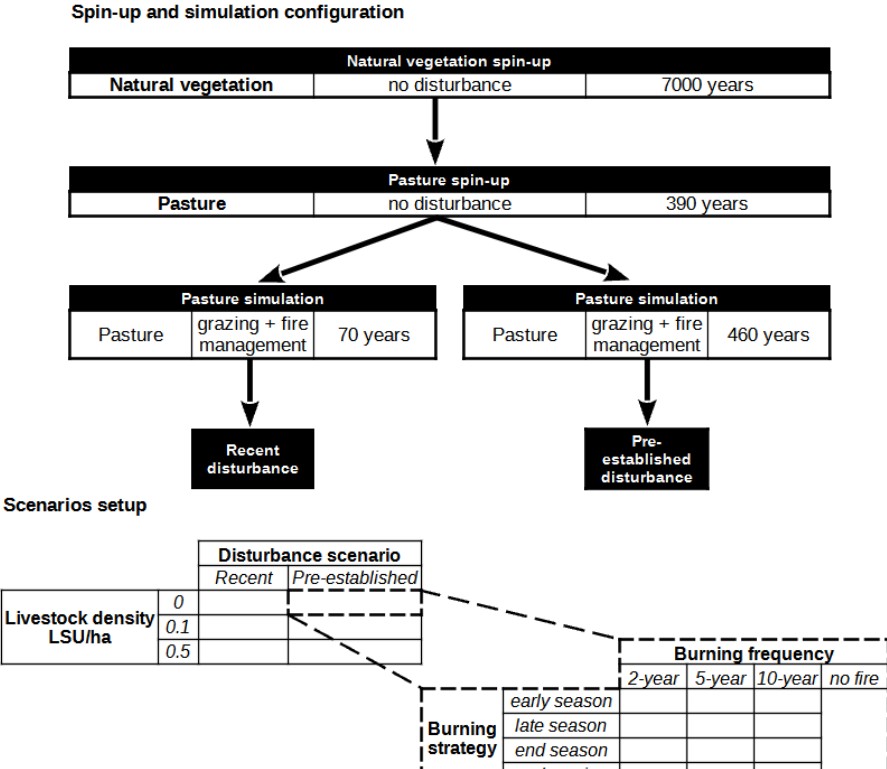

**Figure 3.** Scenario setup for the simulations of pasture management with disturbances, livestock densities, and burning strategies calculated by Chalumeau. The upper part portrays the spin-up and simulation setup, depicting how natural vegetation is being transformed into managed pastures under recent and pre-established disturbance scenarios over time. The lower part explains the scenario conditions, including disturbances, livestock densities, and burning strategies with different seasonal timings and frequencies.

5, to 10 years, with each frequency scenario replicated based on the starting year. For instance, the 2 year burning frequency scenario is executed twice: once with burning beginning in year 0 and once in year 1. In order to simplify and limit the number of scenarios, the burned fraction is set to 1 assuming complete burning of the pasture stand. The final aspect of the management scenario involved the burning date, determined by the four strategies calculated using the Chalumeau algorithm (Sec. *2.2.1*).

## 2.4   Post-processing

LPJmL can generate multiple yearly, monthly, and daily outputs to describe the evolution of the biosphere. This study explores the impact of fire practices on vegetation conditions, with a particular focus on their contribution to the soil fertilisation. The above-ground biomass (AGB) is assessed through the annual leaf biomass and the field productivity by the dry matter intake of the livestock. The recovery status after burning is evaluated using the cumulative Net Primary Production (NPP) over one post-fire month. The NPP represents the carbon assimilation from the atmosphere to the plant and gives an indication of the





regrowth process. Observing the soil C:N ratio and the evolution of SOM-N and SMM-N pools can provide insights into how soil nitrogen is affected by fire practices and help detect potential fertilisation effects. A detailed analysis of the nitrogen cycle fluxes – such as biological nitrogen fixation (BNF), atmospheric deposition, leaching, denitrification, volatilisation, and plant uptake – may enhance understanding of the key dynamics underlying the interaction between fire practices and soil nitrogen.

As explained in the model configuration section 2.3.2, four main burning-frequency scenarios are examined, each with distinct starting years. In order to facilitate comparison between the scenarios, we averaged output variables over all scenarios with the same burning frequency.

### 2.4.1 Exclusion of locations and scenarios

Under certain conditions, applying specific scenarios leads to a short-term state of the grass biomass that is insufficient to sustain livestock. A viability threshold is established and set at 80% of the dry matter intake requirement, which is commonly fixed to 356 g DM m$^{-2}$ annually for 1 LSU ha$^{-1}$ (Rolinski et al., 2018), adjusted according to livestock density. Since burning practices are closely linked to livestock activity, it would be unreasonable to retain scenarios where burning renders the pasture insufficiently productive to sustain animal feeding. Therefore, during the analysis, scenarios where the averaged dry matter intake over 70 years of core simulation phase falls below this threshold are excluded.

### 2.4.2 Normalisation of C:N ratio

The leaf or soil C:N ratio is calculated as the ratio of organic carbon to nitrogen in the corresponding pool. To isolate the impact of disturbances and enable comparisons between sites, the C:N ratios are normalised using the reference C:N ratio of each site under undisturbed conditions (i.e. without burning or grazing).

The results are then expressed as the multi-year average percentage change relative to the reference scenario, following the formula (Eq. 2).

$$\text{Normalised C:N ratio} = \left( \frac{\text{C:N ratio}}{\text{C:N ratio reference}} - 1 \right) \times 100 \tag{2}$$

### 2.4.3 Nitrogen in- and out-fluxes

LPJmL provides multiple nitrogen outputs that together describe the ecosystem and soil nitrogen budget. The ecosystem nitrogen budget is determined by the balance of nitrogen input —— biological nitrogen fixation (BNF) and atmospheric deposition — and nitrogen outputs, including leaching, denitrification, volatilisation, plant uptake, harvest nitrogen (in grazing systems) and NO$_X$ emissions from fire (in burning scenarios). The soil nitrogen budget accounts for nitrogen fluxes associated with the SOM-N and SMM-N pools. Inputs include litter fall, atmospheric deposition, BNF, manure (in grazing systems), and ashes from fire (in burning scenarios). Outputs consist of leaching, nitrification, denitrification, volatilisation, and plant uptake.





# 3 Results

## 3.1 Vegetation condition and field productivity


The interaction between fire practices, grazing, and vegetation dynamics is very important in the evaluation of the productivity and the balance of grassland ecosystems. Both burning practices and livestock density affect the carbon and nitrogen content of the vegetation which, in turn, has consequences for the productivity of the field. This section focuses on the various effects of these practices on the different types and levels of vegetation cover, nitrogen supply, and agricultural productivity at distinct

locations in Brazil, representatively for the Cerrado and the Pampas sites.

### 3.1.1 Above-ground biomass decline and lower nutritional supply

Long-term burning practices, represented by the pre-establish disturbance scenario, lead to a drop in leaf carbon between 30% and 85% compared to an undisturbed condition value of 140 gC m$^{-2}$ without fire and grazing (Fig. 4, b). When burning practices are combined with a livestock density of 0.1 LSU ha$^{-1}$ (Fig. 4, d), the range is smaller with reductions between 75%

and 88%. The overall decrease in vegetation becomes more pronounced with higher burning frequencies.

Earlier burning dates have a negative critical impact on AGB development. The difference between 'early season' and 'early spring' burning strategies is, on average, larger than 30% (Fig. 4). The recovery status appears to be largely driven by the burning timing. Later burning dates show higher cumulative net primary productivity (NPP), suggesting a faster post-burning recovery.

Other regions, such as the South Atlantic Forest and the area of Caatinga, exhibit similar patterns (Fig. B1). In the Pampas site, however, later burning practices lead to the lowest vegetation levels (Fig. 5) with an average deficit of 16% as compared to the 'early season' strategy.

Burning practices exacerbate the nitrogen deficit in leaves, significantly affecting the nutrient balance of the vegetation. Within pre-established disturbance scenario, the leaf C:N ratio strongly increases, between 36% and 70% at the Cerrado and

between 13% and 24% at the Pampas sites, depending on the burning frequency (Fig. 6).

### 3.1.2 Impact on dry matter intake

The dry matter intake is directly derived from the leaf carbon dynamics. In scenarios where the leaf carbon pool is significantly impacted, e.g. under higher frequency and earlier burning strategies, the dry matter intake is also affected. At the Cerrado site, in the recent disturbance scenarios with a livestock density of 0.1 LSU ha$^{-1}$, 2 year burning frequency and 'early season'

burning (text written in white letters in Fig. 7), the dry matter intake decreases down to 25% falling below the viability threshold of 28.49 g m$^{-2}$, i.e. the productivity is insufficient to feed the animals.

Contrary to the leaf carbon pool, which decreases over time represented by the difference between the recent and pre-established disturbance scenarios (Fig. 4), the dry matter intake declines at the introduction of the disturbance (Fig. 7, a) to





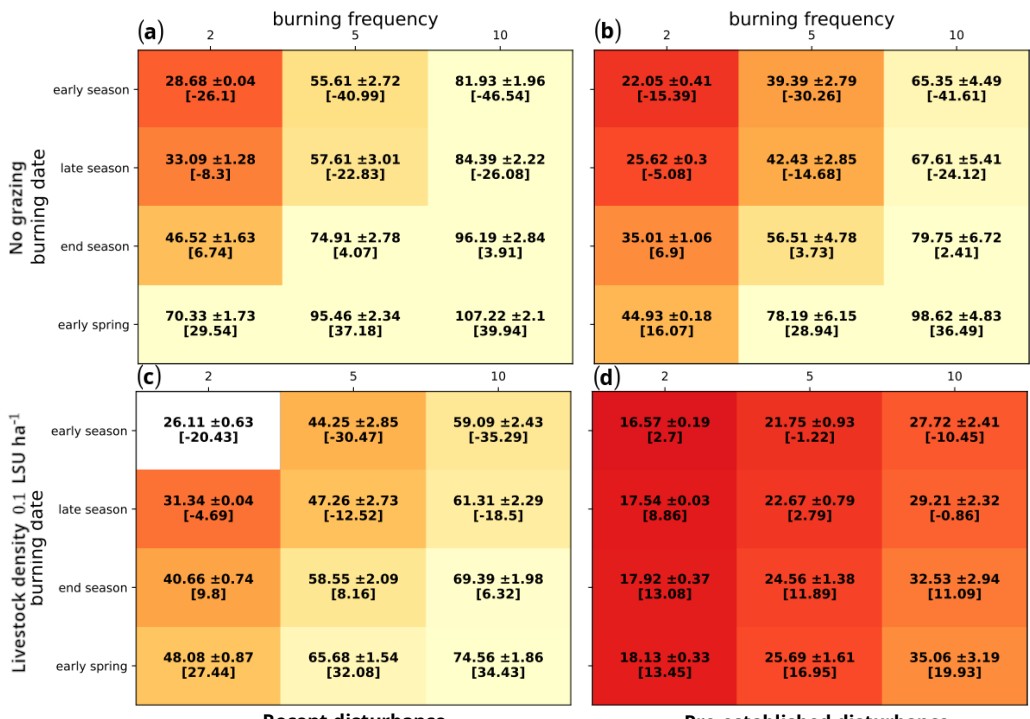

**Figure 4.** Aggregated average leaf carbon (mean ± standard deviation) over a 70-year period on recent or pre-establish disturbance scenarios, displayed by frequency (columns) and burning date (rows), at the Cerrado site. Under undisturbed conditions, leaf carbon reaches 140 gC m$^{-2}$. The 1-month post-fire average cumulative NPP, shown in brackets, provides insights into the recovery status. The colouring represents the magnitude of the reduction in leaf carbon compared to undisturbed conditions, with light yellow indicating a smaller impact and red indicating a significant one. White spaces within the mosaics represent non-selected scenarios due to insufficient dry matter intake fulfilment (Sec. 3.1.2).

a lower value than the viability threshold but stabilises in the pre-established disturbance scenario. 'Early season' burning

strategy constitutes the most affected case with an average dry matter intake decrease of 4%.

Under increased livestock density of 0.5 LSU ha$^{-1}$, the dry matter intake is significantly impacted due to the extremely low vegetation levels in the Cerrado site (Fig. 7, c and d). All scenarios fall below the viability limit of 178 g m$^{-2}$. It is important to note that even without burning practices, this livestock density is too intensive and dry matter intake frequently falls below the viability limit. The average dry matter intake for recent and pre-established grazing are 309 g m$^{-2}$ and 42 g m$^{-2}$, respectively,

which are well below the viability limit.





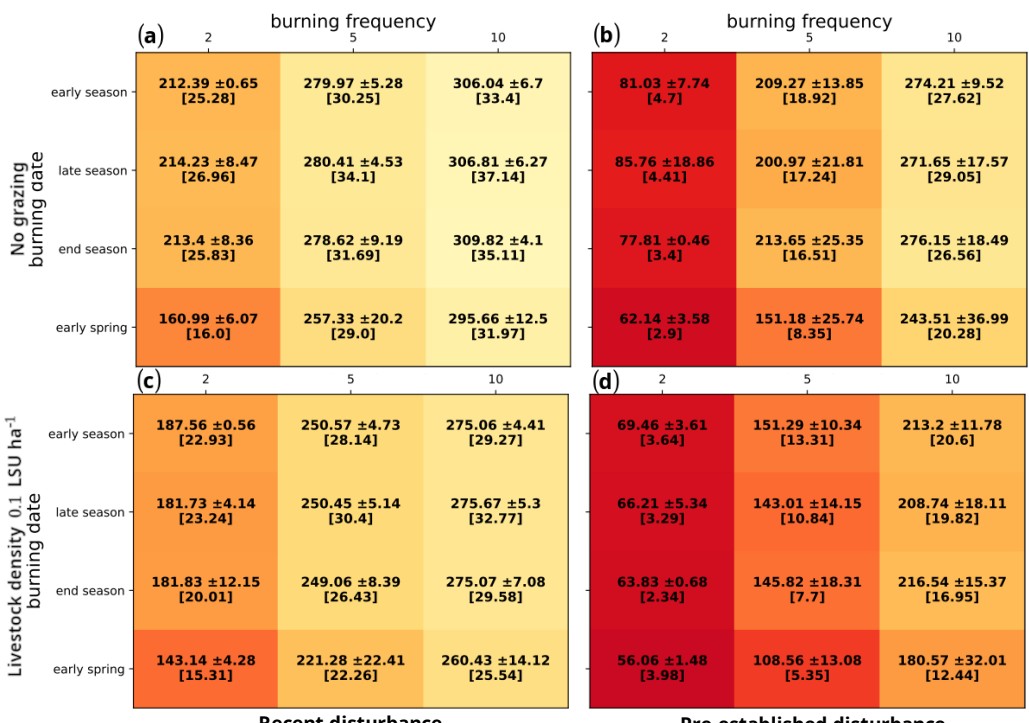

**Figure 5.** Aggregated average leaf carbon (mean ± standard deviation) over a 70-year period on recent or pre-establish disturbance scenarios, displayed by frequency (columns) and burning date (rows), at the Pampas site. Under undisturbed conditions, leaf carbon reaches 330 gC $m^{-2}$. The 1-month post-fire average cumulative NPP, shown in brackets, provides insights into the recovery status. The colouring represents the magnitude of the reduction in leaf carbon compared to undisturbed conditions, with light yellow indicating a smaller impact and red indicating a significant one. White spaces within the mosaics represent non-selected scenarios due to insufficient dry matter intake fulfilment (Sec. 3.1.2).

The drastic grazing impact of 0.5 LSU $ha^{-1}$ is present in the results of all studied sites (Fig. B4). In some regions, such as the Atlantic Forest, Amazon, and Pampas locations, short-term practices remain viable. However, for all sites, long-term practices under intensive livestock density, with or without burning practices, lead to non-viable conditions.

## 3.2 Soil conditions and nitrogen budget

This section puts emphasis on the imbalance of the rapid fire-related changes in fluxes and the slower changes in pools. More specifically, the assessment of the burning practices includes approaches that address the burning frequency, the burning timing,

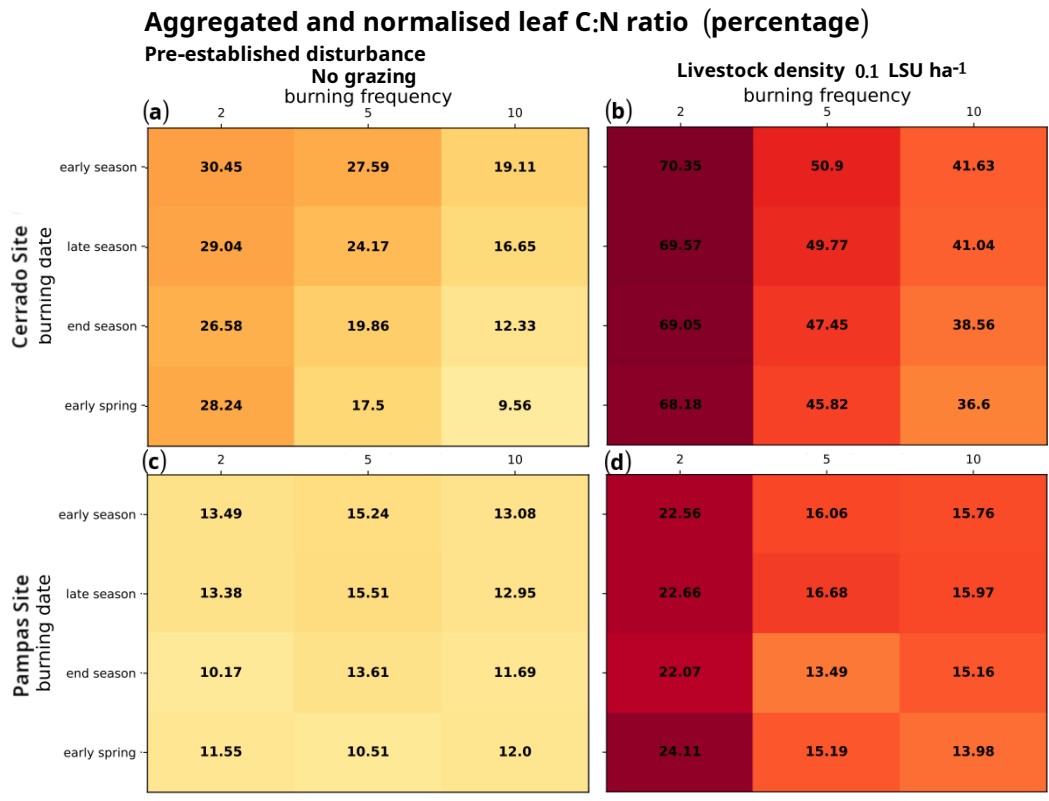

**Figure 6.** Aggregated average and normalised leaf C:N ratio (mean in percentage) over a 70-year period on pre-establish disturbance scenario, displayed by frequency (columns) and burning date (rows), at the Cerrado and Pampas sites. C:N ratios are normalised using the undisturbed scenario as reference, which equal respectively to 15 and 20 (Sec. 2.4.2). The colouring represents the magnitude of the increase of the leaf C:N ratio compared to undisturbed conditions, with light yellow indicating a smaller impact and red indicating a significant one.

and the livestock density, along with their effects on SOM-N and SMM-N pools and the nitrogen budget of the ecosystem and the soil.

### 3.2.1 Soil nitrogen impoverishment

Our results show that the nitrogen deficit increases with the frequency of burning practices. In fact, even in undisturbed scenarios, the soil in the Cerrado site is not rich in nitrogen, as indicated by a soil C:N ratio of 16.65, which is above the optimum value of 15. In the case of the pre-establish disturbance scenario, C:N ratios rise by 1.5% to 4.2% only by burning practices and up to 6.9% in combination with a livestock density of 0.1 LSU ha$^{-1}$ (Fig. 8, b and d). The primary cause of this rise is the unbalanced reduction in both the soil organic carbon and nitrogen pools over time, which is more pronounced for nitrogen and 320 increases the nitrogen debt. However, in the recent disturbance scenario, we notice that the introduction of burning practices



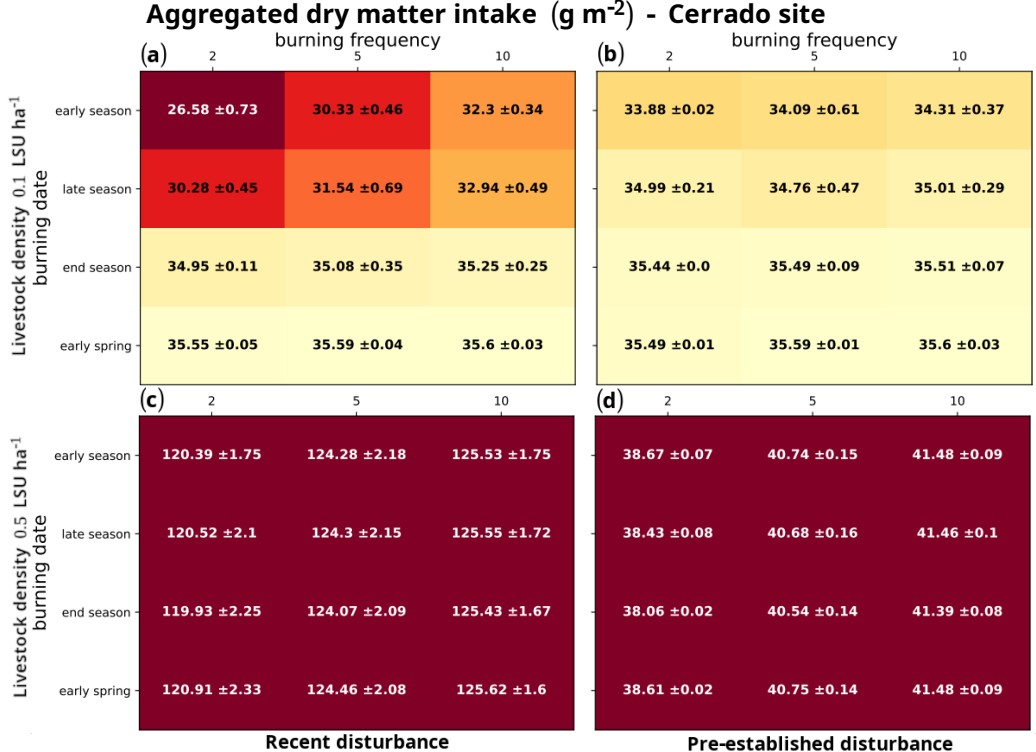

**Figure 7.** Aggregated average dry matter intake (mean $\pm$ standard deviation) over a 70-year period on recent or pre-establish disturbance scenarios, displayed by frequency (columns) and burning date (rows), at the Cerrado site. Nutritional requirements for a livestock density of 0.1 and 0.5 LSU ha$^{-1}$ are respectively 35 and 178 g m$^{-2}$. Under 80% of these values, scenario is considered non-viable and is represented in white text. The colouring represents the magnitude of the reduction of dry matter intake compared to the nutritional requirement, with light yellow indicating a smaller impact, red a significant one and red burgundy representing a fall below the viability threshold.

helps to alleviate the initial nitrogen deficit, decreasing the C:N ratio up to 1.6% without grazing and with frequent burning (Fig. 8, a).

In the Cerrado biome, the nitrogen content in SOM and SMM pools strongly decline under pre-established disturbance scenario with a livestock density of 0.1 LSU ha$^{-1}$. Particularly, SOM-N decreases between 29.5% (Fig. 9, b) and 45.6% while the decreases for SMM-N range from 73.0% to 86.3% (Fig. 9, d).

The frequency of burning is a significant factor to consider for aggravated SOM-N and SMM-N depletion. However, short-term scenarios with recent disturbances show a slight increase in the SMM-N pool. The introduction of burning practices may enhance this pool by up to 55.4% relative to reference scenarios without burning, where it reaches 0.83 gN m$^{-2}$, and with an 0.1 LSU ha$^{-1}$ livestock density (Fig. 9, c).





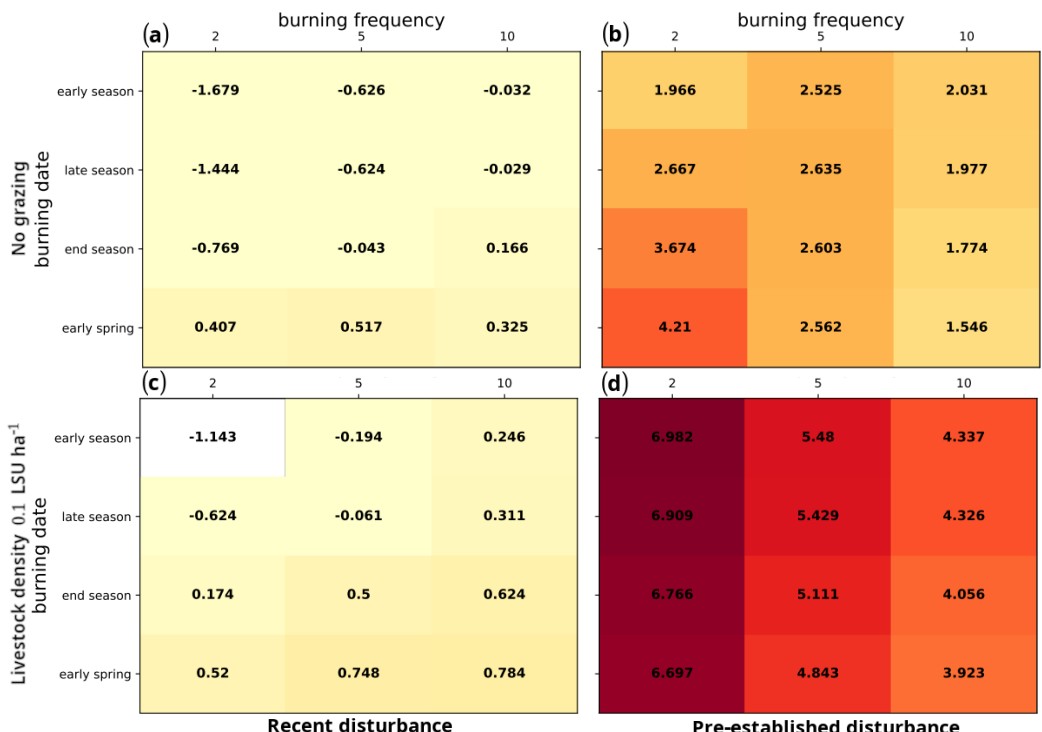

**Figure 8.** Aggregated average and normalised soil C:N ratio (mean in percentage) over a 70-year period on recent or pre-establish disturbance scenarios, displayed by frequency (columns) and burning date (rows), at the Cerrado site. C:N ratio is normalised using the undisturbed scenario as reference, which equals to 16.65 (Sec. 2.4.2). The colouring represents the magnitude of the decrease or increase of the soil C:N ratio compared to undisturbed conditions, with light yellow indicating a smaller impact and red indicating a significant one.

Nevertheless, the timing of burning significantly affects these dynamics. Earlier burning appears beneficial, but a later burning date have the opposite effect. The 'early spring' burning strategy results in a 12.0% decrease in the SMM-N pool compared to reference scenarios with no burning and the same livestock density.

### 3.2.2 Altered nitrogen budgets over disturbance scenarios

Considering the ecosystem nitrogen cycle for the Cerrado site (Fig. 10, a and b), nitrogen inputs, which consist of biological nitrogen fixation (BNF) and atmospheric deposition, decrease with increasing levels of burning activities and grazing. This decline is driven primarily by reductions in BNF, as atmospheric deposition is determined by the inputs provided to the model (Sec. 2.3.1) and, consequently, remains the same regardless of the scenario. The drop is linked to the decline in foliage cover, which is caused by a decrease in plant biomass, leading to a generally lower nitrogen demand. In scenarios involving pre-



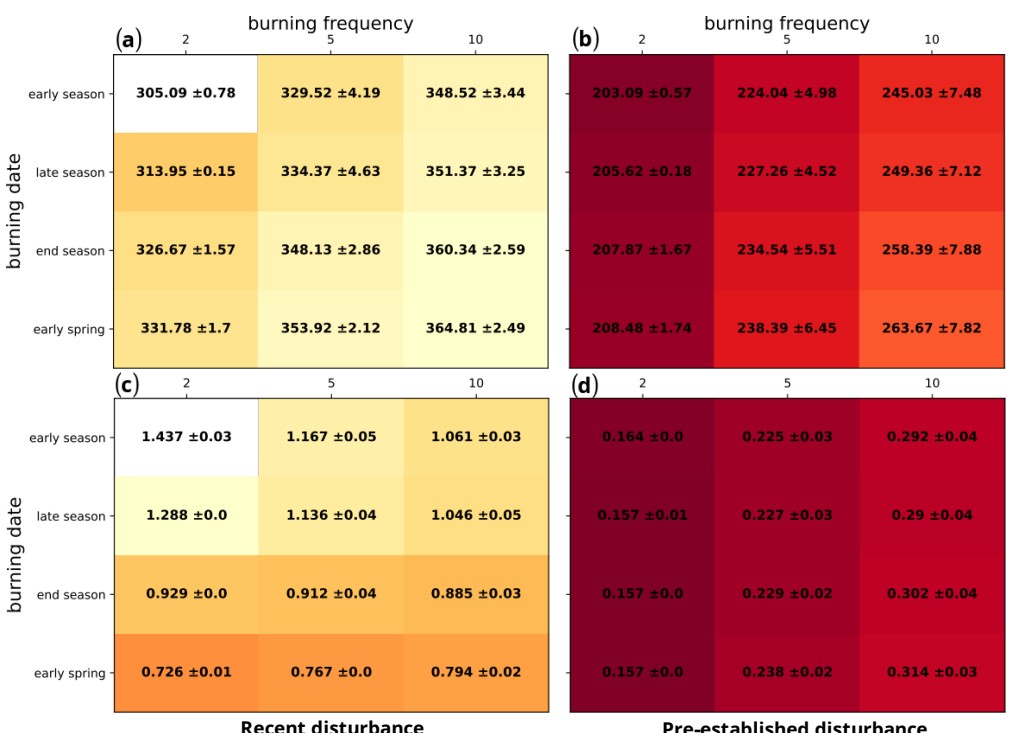

**Figure 9.** Aggregated average SOM-N (a) and SMM-N pools (b) (mean ± standard deviation) over a 70-year period on recent or pre-establish disturbance scenarios, displayed by frequency (columns) and burning date (rows), at the Cerrado site. Under undisturbed conditions, SOM-N and SMM-N are respectively equal to 373.33 and 1.15 gN m$^{-2}$. The colouring represents the magnitude of the reduction in the SOM-N and SMM-N pools compared to the undisturbed conditions, with light yellow indicating a smaller impact and red representing a significant one. White spaces within the mosaics represent non-selected scenarios due to insufficient dry matter intake fulfilment (Sec. 3.1.2).

established and 2 year burning practices, BNF diminishes by up to 60% (Fig. 10b). The Caatinga and the Pampas sites (Fig. 10c

and d) display similar reductions regarding the overall nitrogen input and BNF.

Concerning the nitrogen cycle losses, they are composed mostly of losses from leaching (NO$_3$) and emissions from nitrification and denitrification (N$_2$ and N$_2$O), volatilisation (NH$_3$), harvest N and NO$_X$. When livestock are present, nitrogen removal through dry matter intake has to be considered. In the first 20 years after the introduction of an intensive disturbance, such as high burning frequency or grazing, nitrogen losses are at their maximum. This increase is primarily driven by leaching, which

is proportional to the size of the SMM-N pool. In later years of practice, the dominance of leaching for the losses subsides following the reduction of soil nitrogen pools (Sec. 3.2.1).





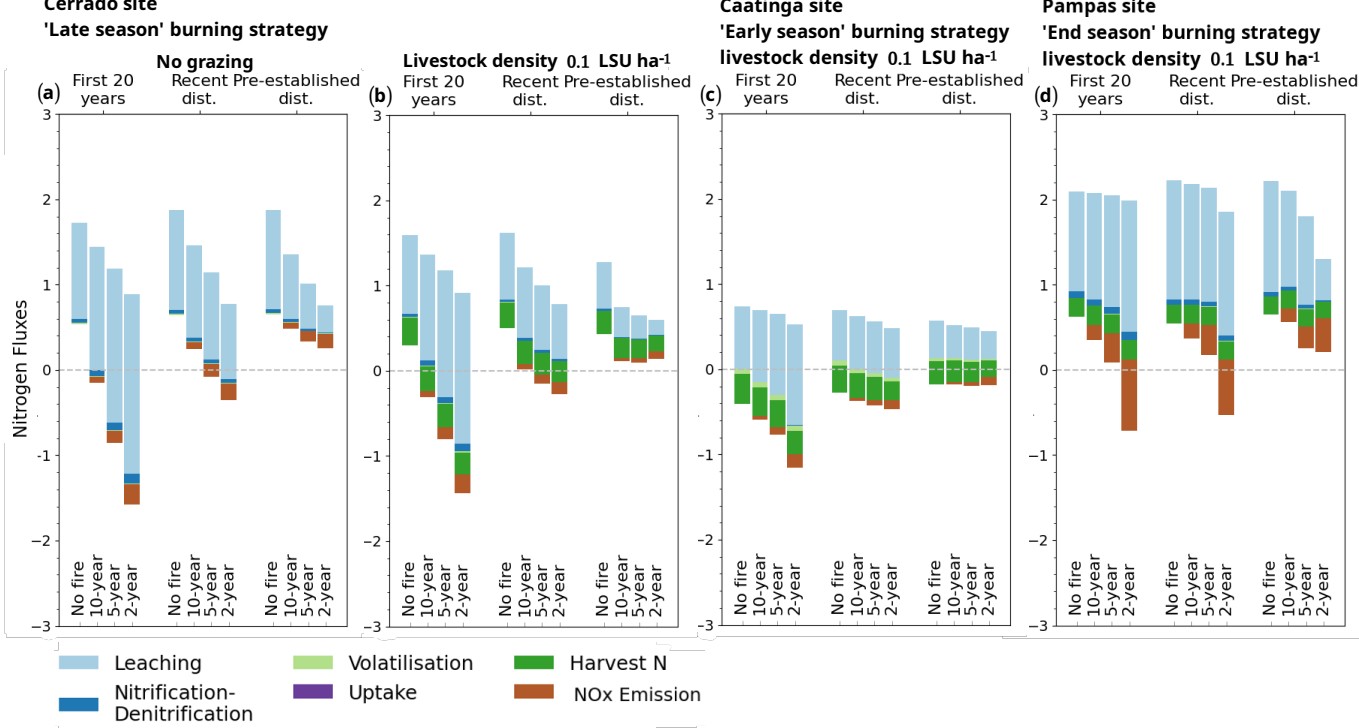

**Figure 10.** Ecosystem nitrogen budget at the Cerrado (a, b), Caatinga (c) and Pampas (d) sites across burning frequencies, livestock densities, and practice durations (Sec. 2.4.3). Nitrogen inputs, consisting of the sum of biological nitrogen fixation (BNF) and atmospheric deposition, are visually represented as the top of the bars in the figure. For clarity, only one burning strategy is depicted for each site, illustrating the observed practices as detailed by Brunel et al. (2021) respectively for the Cerrado, Caatinga and Pampas sites 'late season', 'early season' and 'end season'.

In the first 20 years of disturbances, the overall nitrogen budget is significantly affected, even hitting negative values under scenarios with higher burning frequencies. But as time progresses, the nitrogen fluxes approach an equilibrium state. The introduction of disturbances into the system induces a drastic shift in the nitrogen fluxes, which is the primary driver of the

decrease in nitrogen pools (Fig. 9).

In the case of the soil nitrogen budget (Fig. 11, a and b), only BNF and litterfall — which is directly attributable to the plant carbon and nitrogen pools – are the primary input fluxes of nitrogen. As stated in section 3.1.1, with increasing intensity of these practices gradually, and over time, the leaf pools decrease, which causes a reduction in litterfall. This drop also affects the nitrogen uptake of plants from the soil SMM-N pool, which in turn are affected by the same fate.

In the Caatinga region (Fig. 11, c), extreme water stress limits vegetation growth, which invariably leads to a negative nitrogen budget. Even without fire practices, the ecosystem experiences a net nitrogen loss primarily due to minimal grazing. As earlier mentioned with regard to the Cerrado, disturbance factors like burning frequency or livestock activity intensification





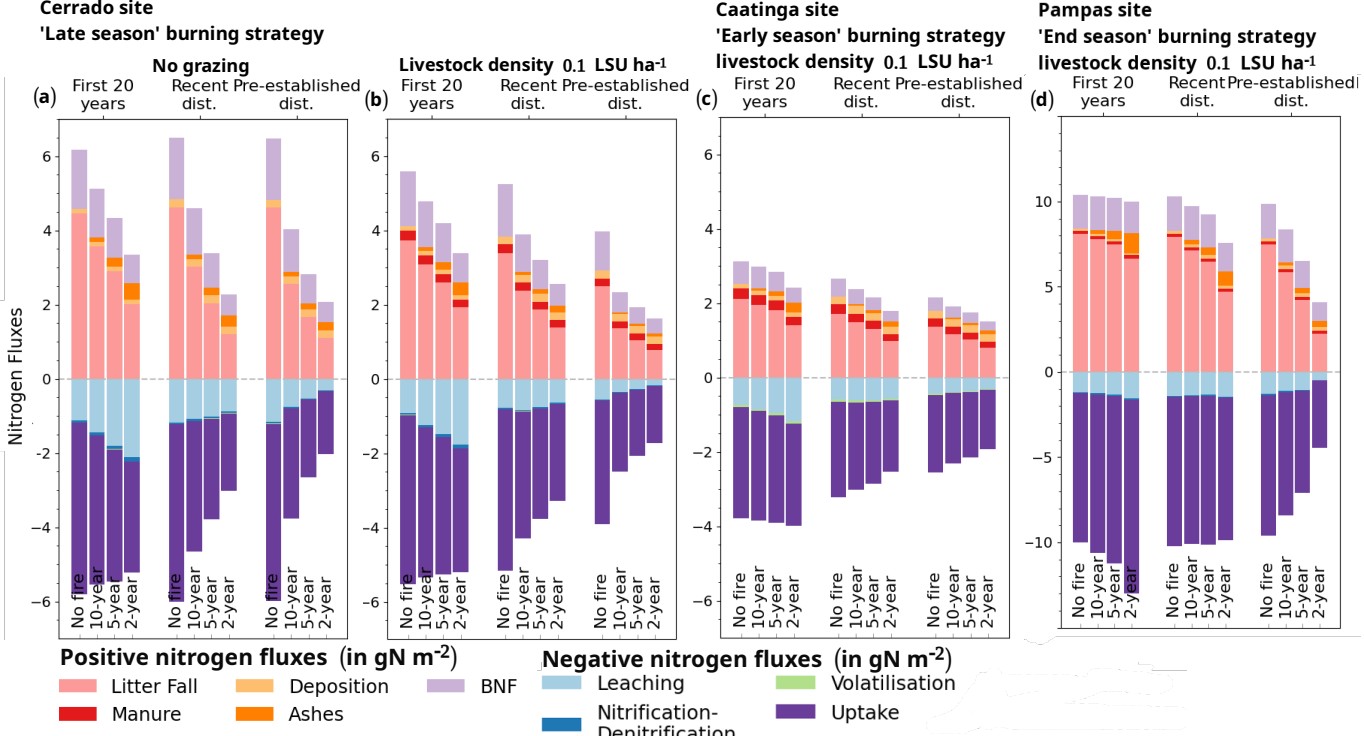

**Figure 11.** Soil nitrogen budget at the Cerrado (a, b), Caatinga (c) and Pampas (d) sites across burning frequencies, livestock densities, and practice durations (Sec. 2.4.3). The nitrogen budget summarises the N-input and output fluxes coming in and out from SOM-N and SMM-N pools. For clarity, only one burning strategy is depicted for each site, illustrating the observed practices as detailed by Brunel et al. (2021) respectively for the Cerrado, Caatinga and Pampas sites 'late season', 'early season' and 'end season'.

lead to even higher nitrogen losses. Also, at the Pampas site (Fig. 11, d), the nitrogen budget can be summarised by a steady decrease of nitrogen inputs, which is the norm for the nitrogen budget. However, it does not reach negative values except for 360 the 2 year fire frequency in a short-term period.

# 4 Discussion

This study examines the impact of fire practices on grassland ecosystems using the Chalumeau algorithm integrated into the SPITFIRE module of the LPJmL DGVM. Our findings show a substantial reduction in the overall ecosystem and soil nitrogen budget with repeated burning, indicating that frequent fires can degrade soil fertility over time. The depletion of soil 365 nitrogen, along with a decrease in the soil carbon pool, negatively affects vegetation, leading to a decline in pasture productivity, especially under intensive grazing. However, with moderate livestock density and across nearly all fire regime tested, soil and vegetation pools reach a sustainable equilibrium, maintaining adequate dry matter intake and supporting long-term viability.





Notably, in LPJmL, it is assumed that burning itself does not directly alter the leaf C:N ratio, as both carbon and nitrogen pools are reduced proportionally during fire events in the SPITFIRE module. The difference occurs during the regrowth phase,
when the balance of allocation shifts from carbon to nitrogen dominance. Over time, the total amount of nitrogen assimilated by the plants decreases compared to carbon, resulting in a more severe nitrogen deficit.

Grazing and fire practices, applied separately or combined have a significant influence on the above-ground biomass net primary productivity (ANPP). Walker (1999) found that the combined effects of both practices are most beneficial in humid regions, observing it across multiple grassland sites in the US. This indicates that precipitation is needed to achieve high
productivity to balance vegetation loss from fire. Our study covers multiple locations in Brazil that do not match the humid climate described by Walker (1999). The Caatinga and the Cerrado regions, for instance, are characterised by dry climatic conditions. For both sites, the joint application of both practices negatively impacts the ANPP as shown by our result regarding the vegetation level and the dry matter intake. Conversely, in the Pampas, which experiences a wetter climate with year-round rainfall, the effects of grazing and fire on productivity are more favourably compared to other regions. This highlights how a
site-specific precipitation regime influences the response of ecosystems to these disturbances: in wetter regions like the Pampas, fire and grazing can coexist with higher vegetation productivity, whereas in drier areas like the Cerrado and the Caatinga, these disturbances severely impact pasture health.

Burning timing appears from our results to be another parameter linked to climate conditions. Earlier burning strategies critically impact the AGB development as seen in the section 3.1.1, as burning during the dormant season, when growth is
inactive, hinders post-fire recovery. An exception to this context is observed in the Pampas, where burning earlier during winter leads to higher vegetation levels and a slightly more efficient recovery compared to other sites. This behaviour is due to the region's seasonal temperature pattern, which supports the vegetation cycle during the dormant period. In this area, the vegetation's regrowth slows down over the winter, resuming later in time to benefit from the warmer summer. In this way, burning at the end of winter tends to deteriorate conditions for optimal vegetation regrowth, whereas burning earlier in winter
has minimal impact on the natural vegetation cycle.

As noted in the section 3.2.1, the soil is subject to a nitrogen impoverishment, and the soil nitrogen budget (Sec. 3.2.2) shows a substantial drop in the nitrogen uptake over time when fire and grazing practices are intensified. Additionally, a reduction in the soil carbon pool occurs (Sec. A), driven by the decrease in AGB and, consequently, in the primary input into the soil by shed leaves. These findings align with the observations made by Ojima et al. (1994), who investigate the short-term effects of
fire on production and microbial activity in the tallgrass prairie in Kansas (US). They also examine the long-term consequences of annual burning on SOM and nutrient cycling through a combination of the field, laboratory, and modelling studies. Their research reveals significant reductions in SOM-N, microbial biomass, nitrogen availability, and an increase in the C:N ratio of SOM following fire. Using the CENTURY model, they simulate a decrease in soil carbon and net nitrogen mineralisation.

One important direct outcome of fire is ash production, which is believed to enhance soil fertility (Alencar et al., 2011;
Barlow et al., 2020; Pivello et al., 2021). Our result shows a beneficial short-term effect after the introduction of fire expressed as a slight decline of the soil C:N ratio and an increase of the SMM-N pool by up to 50%, which suggests a brief enhancement of the soil nitrogen pool. Looking at sub-annual dynamics (not shown), we observe that this enhancement is due to the input of




nitrogen through ash in a few days period, thereby reducing the soil C:N ratio. Nitrogen from ashes contributes to the SMM-N pool and is subsequently immobilised when the soil C:N ratio is above its optimum, which is, in general, the case at all our
study sites. Consequently, the C:N ratio shows a slight decrease.

Finally, an important aspect and especially relevant in the context of livestock farming is the impact of fire practices on the grassland production. In their observation and experimentation with the CENTURY model, Ojima et al. (1994) noticed a minimal impact on grass production. In our simulations, dry matter intake is considered as an indicator of productivity. Our results indicate that under moderate livestock density, there is an initial reduction in intake, but it balances over time, maintaining
approximately 80% of the livestock's feed requirement (Sec. 3.1.2). This dynamic differs from the decrease observed for the AGB (Sec. 3.1.1). With the pre-established disturbance scenario, so after long-term application of disturbances, we obtain a system with less biomass but a stabilised intake. The differences in how pools and fluxes respond to disturbances drive these outcomes. In this example, the AGB pool is originally not affected much since grazing and burning only reduce the carbon pool by a small share. Over time, however, an incremental decline becomes apparent, leading to a drastic reduction in biomass
by the end of the pre-established disturbance scenario. For the dry matter intake, the picture looks different since magnitudes of the net flux and loss from burning are comparable (Sec. 3.1.2). The intake is initially strongly impacted by the disturbances, which diminish over time as biomass decreases. This leads to a gradual return of the intake into a stable equilibrium, being less affected by the shrinking disturbance intensity due to the lack of fuel availability. This balance can be maintained as long as the biomass remains high enough to supply sufficient intake.

However, increasing grazing intensity leads to a collapse in pasture viability, rendering it unsustainable for livestock. From our results we cannot conclude about the viability of livestock densities above 0.5 LSU since we did not include fertilisation in our stylised scenarios.

While the results represent a great advancement and novelty in how fire management practices are modelled in DGVMs, it is important to recognise that many limitations are still present. By thoroughly identifying and discussing these critical points,
we aim to provide a foundation for enhancing the accuracy and applicability of future models in simulating fire impacts on grassland's vegetation and soil.

In this study, we investigate the effects of burning practices on pasture ecosystems using model simulations for a set of burning and grazing scenarios. The proposed scenarios are comprehensive but do not account for all real field conditions.

For example, the burned area is prescribed as an input parameter. For burning dynamics to be properly assessed, the most
relevant method is to perform burning of the entire pasture, in other words, the burned area parameter is set to 1 and remains constant throughout the entire period. In reality, farmers might conduct rotational burning, selecting fire spots based on their own observations regarding the field status (Pivello et al., 2021). Criteria such as the amount of dead biomass, small bushes, or toxic plants for cattle often guide these decisions (Pivello, 2011; López-Mársico et al., 2019), a dimension not represented in LPJmL. In the LPJmL grass modelling approach, there is no distinction between living and dead biomass within the plants.
During pasture burning, all plant biomass is treated as fuel, although the fire will affect primarily moribund plant parts, particularly when fire occurs at the end of the dormant season. This timing is critical considering that the dead biomass build-up takes place in this period, which entices farmers to burn it. Implementing a proper way to separate the living and dead biomass



would enable a full usage of SPITFIRE functionality specially the one regarding fuel estimation and fire ignition condition. Incorporating this aspect into the determination of the burning date, in addition to the already existing climate condition, might

better align with the initial purpose of setting fire.

## 5    Conclusions

This study investigates the impact of pasture fire practices on vegetation conditions, field productivity, and soil fertility in grasslands. This is achieved by integrating the Chalumeau algorithm into the SPITFIRE module within the LPJmL DGVM and performing a set of sensitivity simulations according to a handful of management scenarios.

We demonstrate the negative effects of intensive and prolonged combined disturbances, such as grazing and burning, on grassland ecosystems. Indeed intensive grazing alone can drastically impact the vegetation and soil carbon and nitrogen pools, leading to non-viable conditions for livestock rearing. When combined with burning as a biomass management strategy, this degradation is further amplified. However, our results indicate that parsimonious burning, in accordance with livestock density, can establish a new equilibrium, retaining the ecosystem sustainable for livestock. Additionally, the impacts vary significantly

depending on the climatic conditions, with wetter climates exhibiting greater resilience compared to drier areas.

This background should be taken into consideration when attempting to evaluate current land management practices in Brazilian pastures. To conduct such evaluations effectively, it is essential to obtain context-specific knowledge about actual practices, such as burning frequency, timing, fertilisation application, and the extent of burned areas. This highlights the need for the scientific community to broaden their appreciation of fire practices by better methods of data collection, monitoring of

grasslands and further investigations of the issues raised in this paper.

*Code and data availability.* The LPJmL5 Pasture-Burning version used to produce the results of this paper as well as the data and the post-processing python script are archived on Zenodo at DOI: https://doi.org/10.5281/zenodo.14926359





## Appendix A: Soil carbon decline driven by fire frequency

This section examines the decline in soil carbon over time, exacerbated by fire disturbances. The results indicate that higher fire
frequencies lead to more pronounced declines in soil carbon. At both study sites, soil carbon levels under disturbed conditions
show reductions of up to 45% compared to undisturbed conditions under pre-established disturbance scenarios.

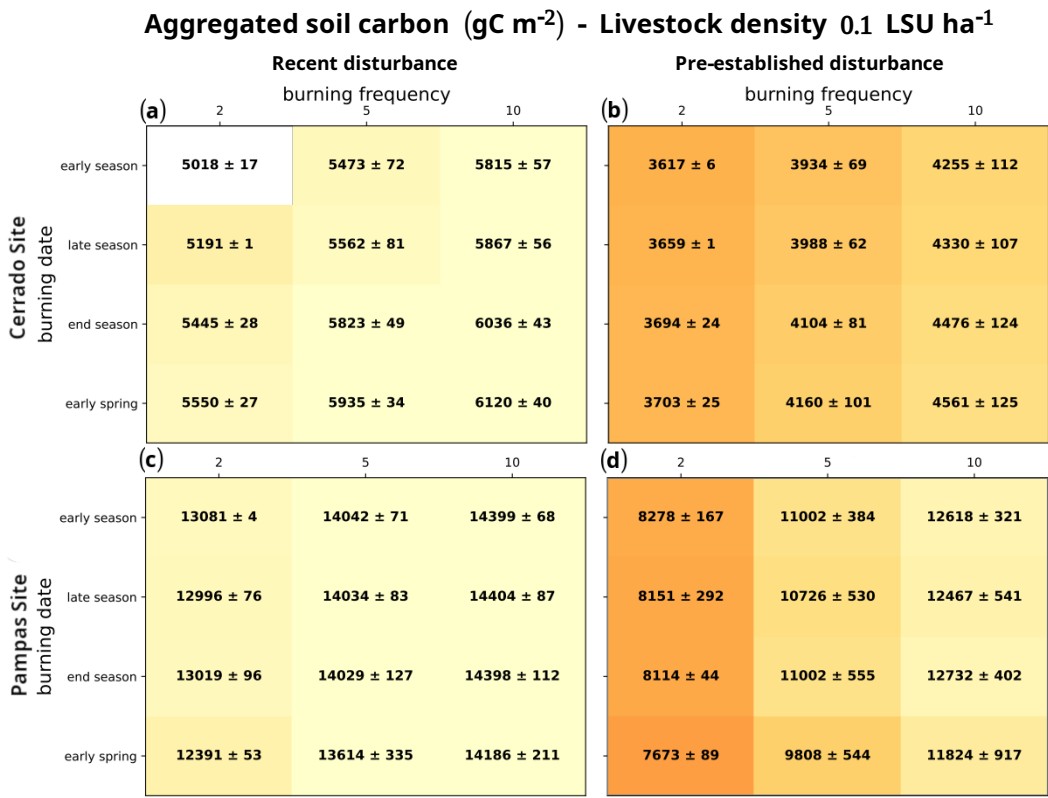

**Figure A1.** Aggregated average soil carbon (mean $\pm$ standard deviation) over a 70 year period on recent or pre-establish disturbance scenarios, displayed by frequency (columns) and burning date (rows), at the Cerrado (a and b) and Pampas (c and d) sites. Under undisturbed conditions, soil carbon is equal respectively to 6220 and 14670 gC m$^{-2}$. The colouring represents the magnitude of the reduction in soil carbon compared to undisturbed conditions, with light yellow indicating a smaller impact and red indicating a significant one. White spaces within the mosaics represent non-selected scenarios due to insufficient dry matter intake fulfilment (Sec. 3.1.2).



**Appendix B: Additional results for DMI, leaf C:N ratio, and nitrogen budget**



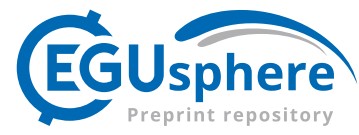

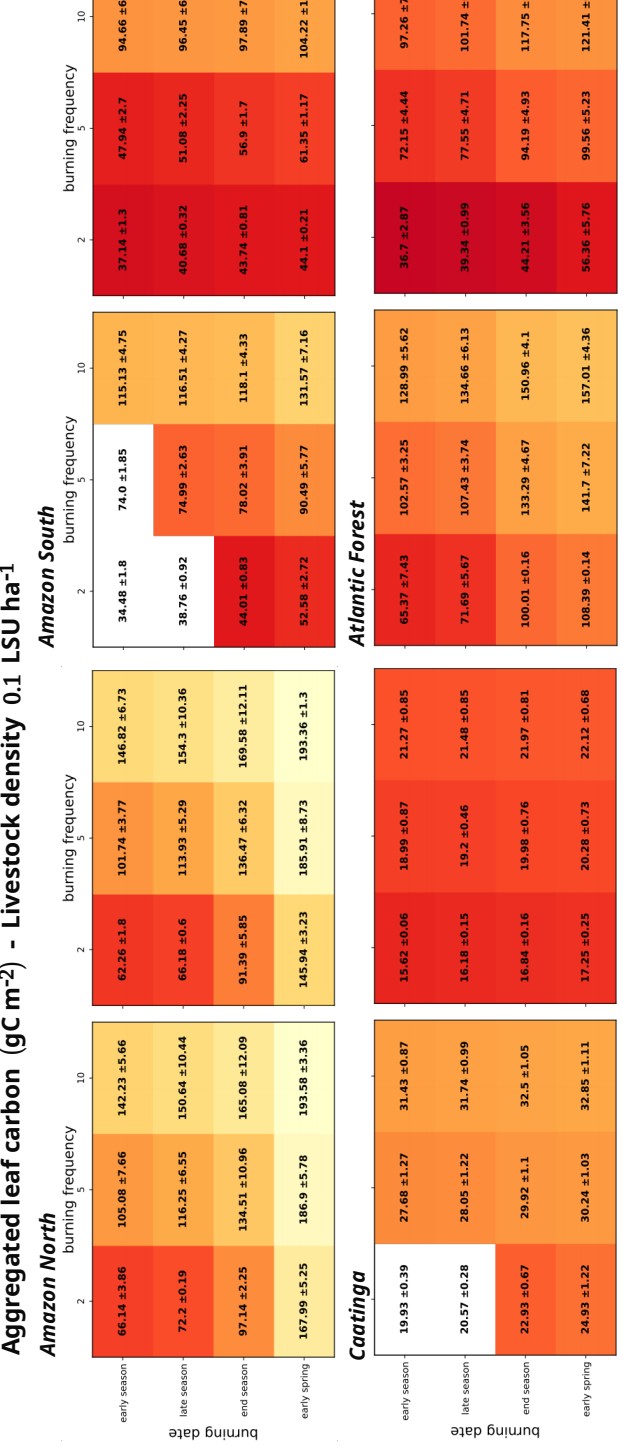

**Figure B1.** Aggregated average leaf carbon (mean ± standard deviation) over a 70 year period on recent or pre-establish disturbance scenarios, displayed by frequency (columns) and burning date (rows), at the Amazon North and South, the Caatinga, the Atlantic Forest sites under livestock density equals to 0.1 LSU ha$^{-1}$. Under undisturbed conditions, leaf carbon reaches accordingly 180, 175, 40 and 230 gC m$^{-2}$. The colouring represents the magnitude of the reduction in leaf carbon compared to undisturbed conditions, with light yellow indicating a smaller impact and red indicating a significant one. White spaces within the mosaics represent non-selected scenarios due to insufficient dry matter intake fulfilment (Sec. 3.1.2).

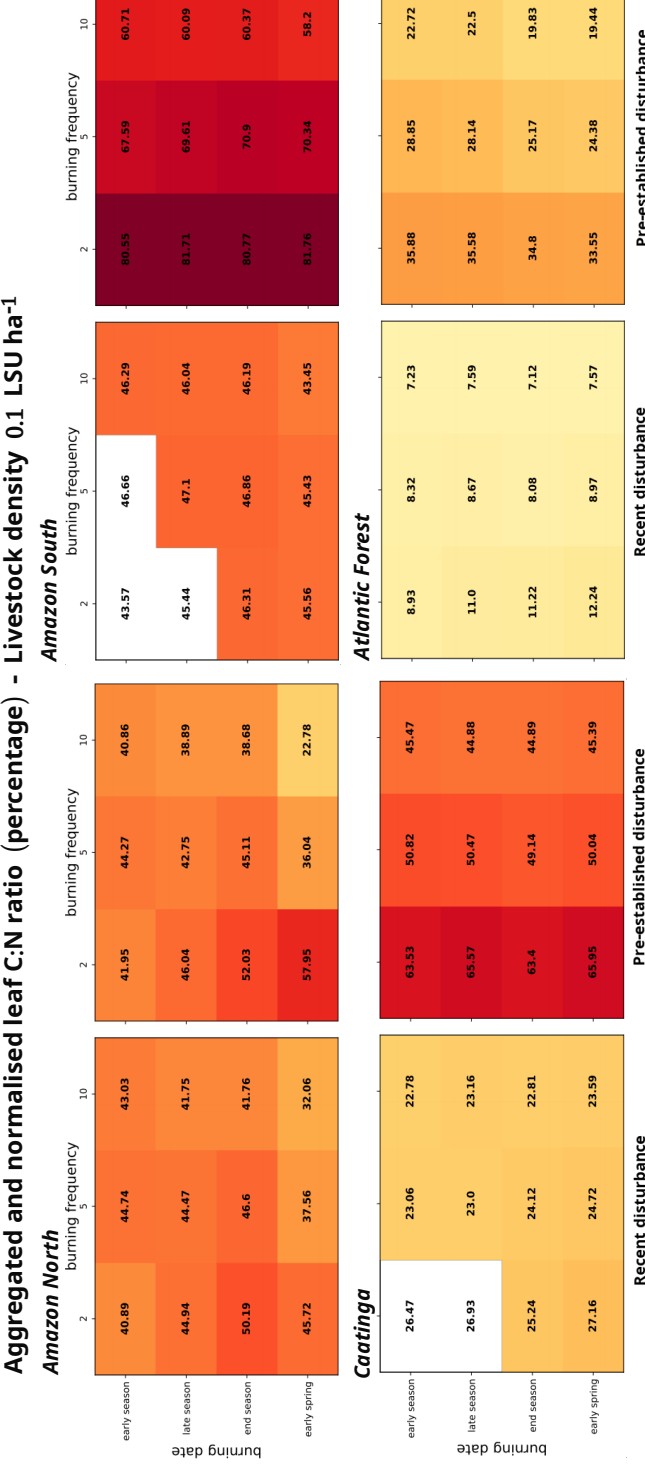

**Figure B2.** Aggregated average and normalised leaf C:N ratio (mean in percentage) over a 70 year period on recent or pre-establish disturbance scenarios, displayed by frequency (columns) and burning date (rows), at the Amazon North and South, the Caatinga and the Atlantic Forest sites under livestock density equals to 0.1 LSU ha$^{-1}$. C:N ratio is normalised using the undisturbed scenario as reference, which equal respectively to 11.4, 11.3, 12.2 and 18.4 (Sec. 2.4.2). The colouring represents the magnitude of the increase of the leaf C:N ratio compared to undisturbed conditions, with light yellow indicating a smaller impact and red indicating a significant one.




**Figure B3.** Aggregated average dry matter intake (mean ± standard deviation) over a 70 year period on recent or pre-establish disturbance scenarios, displayed by frequency (columns) and burning date (rows), at the Amazon North and South, the Caatinga, the Atlantic Forest sites under livestock density equals to 0.1 LSU ha$^{-1}$. Nutritional requirement is 35.61 g m$^{-2}$. Under 80% of these values, scenario is considered non-viable and is represented in white text. The colouring represents the magnitude of the reduction of dry matter intake compared to the nutritional requirement, with light yellow indicating a smaller impact, red a significant one and red burgundy representing a fall bellow the viability threshold.





**Figure B4.** Aggregated average dry matter intake (mean ± standard deviation) over a 70 year period on recent or pre-establish disturbance scenarios, displayed by frequency (columns) and burning date (rows), at the Amazon North and South, the Caatinga, the Atlantic Forest sites under livestock density equals to 0.5 LSU ha$^{-1}$. Nutritional requirement is 178.05 g m$^{-2}$. Under 80% of these values, scenario is considered non-viable and is represented in white text. The colouring represents the magnitude of the reduction of dry matter intake compared to the nutritional requirement, with light yellow indicating a smaller impact, red a significant one and red burgundy representing a fall bellow the viability threshold.



**Figure B5.** Ecosystem nitrogen budget at the Amazon North and South and the Atlantic Forest sites under livestock density equals to 0.1 LSU ha$^{-1}$, across burning frequencies and practice duration. For clarity, only one burning strategy is depicted for each site, representing the observed practices as detailed by Brunel et al. (2021) respectively 'early season' for the Amazon sites and 'late season' for the Atlantic Forest.





**Figure B6.** Soil nitrogen budget at the Amazon North and South and the Atlantic Forest sites under livestock density equals to 0.1 LSU ha$^{-1}$, across burning frequencies and practice duration. For clarity, only one burning strategy is depicted for each site, representing the observed practices as detailed by Brunel et al. (2021) respectively 'early season' for the Amazon sites and 'late season' for the Atlantic Forest.



*Author contributions.* MB and SR led the conceptualisation and development of the methodology. MB implemented the computer code with contributions from SR, MD, and KT. MB conducted the formal computational analysis, created the visualisations and prepared the original draft of the manuscript. SR provided supervision throughout the project. The manuscript was reviewed and edited by SR, SW, MD, KT, HB and JH.

*Competing interests.* At least one of the (co-)authors is a member of the editorial board of Biogeosciences.

*Acknowledgements.* The authors gratefully acknowledge the European Regional Development Fund (ERDF), the German Federal Ministry of Education and Research, and the Land Brandenburg for supporting this project by providing resources on the high-performance computing system at the Potsdam Institute for Climate Impact Research. AI algorithms were used during the writing process to assist with English spelling, formulation, and syntax.





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
