# Peer review of "Effects of fire and grazing on biogeochemical cycles in Brazilian pastures using LPJmL5-Pasture-Burning"

_EGUsphere, 2025_

## Referee Comment (RC1)

**Review: "Effects of fire and grazing on biogeochemical cycles in Brazilian pastures using LPJmL5-Pasture-Burning" (egusphere-2025-922)**

**Overview**

In this manuscript, the authors describe how they implemented an algorithm for pasture burning date into the LPJmL dynamic global vegetation model (DGVM). This sort of development is really important—ranchers in many parts of the world exert a strong influence on the frequency and timing of fire in grasslands, which can make a big difference on ecosystem dynamics. Most DGVMs have no such representation of fire management practices, especially an endogenous one (as opposed to one prescribed from input files). While it seems that other data limitations prevent this feature from being commonly used in general LPJmL runs, this capability is an important first step.

The authors do not just describe the technical capability, however. They also use the updated model to assess the separate and joint impacts of management fire and grazing on ecosystem carbon and nitrogen. The results show that both processes are important to represent in DGVMs.

I'm very happy to see a manuscript like this. The paper is mostly written well, and the figures are mostly good, with interpretations mostly well-supported. However, I have a major methodological concern along with a number of smaller questions and suggestions.

**Major comments**

The main thing I'm concerned about here is how grazed nitrogen is handled in the model. There's no citation in the Methods, as far as I noticed, pointing to information about how LPJmL grazing works at all, actually. N isn't mentioned alongside C at L131-133 as being partially returned to the soil via animal waste after grazing. That initially led me to think it wasn't. Then at L267, "manure (in grazing systems)" is mentioned as one input to the soil. But what fraction of the consumed N? And is it only feces or is it also urine? Then again, in Fig. 10, neither manure nor urine are explicitly mentioned. They might be included in "harvest N" with that color showing *net* harvest N loss, but that's not explained. And finally, Fig. 11 does have manure again.

If a realistic fraction of the grazed N isn't returned to the soil, I have serious doubts about the LPJmL model's fitness for the purpose of analyzing impacts of grazing on soil N and thus leaf C:N ratio too. See Selbie et al. (2015, DOI 10.1016/bs.agron.2014.09.004): "Ruminants excrete as much as 70–95% of the nitrogen (N) they consume."

If the N analyses are kept, some text needs to be added to the Methods describing biological nitrogen fixation (BNF) in LPJmL. Are both symbiotic and asymbiotic BNF represented? How do they work?

**Other comments**

1) I would like to see some text added to the Discussion or Conclusion about what work would be needed in order for this feature to become commonly enabled in LPJmL runs. Is it just livestock density maps (both historical and for future scenarios) that are holding it back?

2) The Appendices are strange. Appendix A has only one figure—why not combine the two Appendices? That figure, Fig. A1, is also the very last to be mentioned in the main text (and only as "Sec. A", not "Fig. A1"); it should thus be last in the Appendix as well. Finally, there are four Appendix figures not mentioned in the main text: Figs. B2–3 and B5–6.

3) All figures except Fig. 1, or at least certain labels in those figures, seem have JPEG artifacts. In most cases these should be replaced with entirely vector-based figures (.eps or .pdf). Failing that, PNG should be used. JPEG should only be used for photos (and don't just convert JPEGs to PNG!). See https://www.biogeosciences.net/submission.html#figurestables for more information.

4) The "matrix"-type figures (Figs. 4–9) need a fair amount of work:
   a. All of these figures should have color bars. It should also be made obvious when subplots share a color bar.
   b. There are many cells in these figures with black text on a dark background. This should be avoided, for instance by adding a white outline or "glow" around all text that overlays a non-white background.
   c. Some of these seem to be true values, while others are changes relative to a baseline. This is hard to keep track of and introduces an extra mental load in interpreting them. Please consider standardizing on one or the other.
   d. In some of these, white represents "excluded because of insufficient biomass for grazing," whereas in others it's burgundy. This should be standardized to burgundy (or even better for colorblind readers, black). I say this because white is confusing: In Figs. 4-5, the lightest color (yellow) is low-impact, the darkest color (dark red) is high-impact, and pure white is the highest impact of all. The color scale goes light-dark-light.
   e. Most of the cells in these matrices represent a "bad" impact, so it makes sense they are represented by a yellow-red color scale. However, some represent a "good" impact—e.g., some treatments in Fig. 8 showing soil N

enrichment. In such cases, they should *not* be on the same color scale, because they are qualitatively different. Something other than yellow-red should thus be used—e.g., blues.

    f.    An extra column should be added to the right side of every matrix giving the results with no fire.

Other comments:

5)    L152: Cite Rothermel.

6)    L157-166 (Sect. 2.2.1): Since the Brunel et al. (2021) and Waha et al. (2012) papers are not open-access, more detail should be given here (or in an Appendix/Supplement) on the Chalumeau algorithm and its implementation. For instance, how often is the burning date updated? Does it use a rolling window to calculate seasonality variables? What's the difference between the "burning date" vs. the strategies (e.g. "early spring")?  Etc.

7)    L190: Would it be accurate to replace "utilised" with "combusted"? If so, please do. If not, please explain what "utilised" means here.

8)    L224-233: Is other fire allowed to happen during these experiments? I.e., are the only ignitions allowed due to intentional pasture management burning, or is the rest of SPITFIRE operating at the same time?

9)    L227-229: It's not until this sentence until I understood what this paragraph was supposed to be describing; until then I was pretty confused. Please move it to the top of the paragraph and edit as needed for flow.

10)    L230-231: I'm not sure I understand this correctly. How many replicates does this result in? 2 + 5 + 10 = 17?

11)    L233: Are the four strategies something that the Chalumeau algorithm produces for each gridcell? Or are they things you switch between for different experiments?

12)    L241-243: "may" is confusing. Is this something you're doing or not?

13)    L250-253: "Since burning practices are closely linked to livestock activity, it would be unreasonable to retain scenarios where burning renders the pasture insufficiently productive to sustain animal feeding. Therefore, during the analysis, scenarios where the averaged dry matter intake over 70 years of core simulation phase falls below this threshold are excluded."

14)    L262-268 (Sect. 2.4.3): Why is this in the "Post-processing" section? It would be more appropriate near Sect. 2.1.3 ("Soil nitrogen pools").

15)    L277: "pre-establish" should be "pre-established".

16)    L276-284: This text needs to mention that it's discussing the Cerrado site specifically.

17) L288: What nitrogen deficit? It hasn't been previously established that NPP is N-limited under any conditions or treatment.

18) L295: "the dry matter intake decreases down to 25% falling below the viability threshold" is hard to understand. Please revise.

19) L297–299: How is this result possible?

20) (L298) Fig. 4d: Why were no scenarios excluded (colored white) due to biomass being too low for grazing?

21) L304:
    a. Why does recently-established grazing have such a higher average dry matter intake?
    b. Are these numbers for the "no burning" treatment specifically?

22) (L311) Fig. 6:
    a. It'd be nice to have the site names in the figure title, as is done for similar figures.
    b. C:N normalized to a percentage feels wrong, I think because it's a result of both the numerator and the denominator. Consider changing to actual values instead of percentage.

23) L316-317: What is this "optimum" value? Optimum for what?

24) L317: Move "only" to after "practices".

25) L321: What "initial nitrogen deficit"? Deficit relative to what?

26) (L329) Fig. 8:
    a. Title and first sentence of caption should say that these numbers represent the *change in* C:N ratio. That's different from all the other such figures.
    b. Cells with negative values (indicating enrichment) should not also be yellow. Maybe a light blue instead.
    c. C:N change as a percentage feels wrong, I think because it can result from either a change in the numerator or the denominator. Consider changing to actual values instead of percentage.
    d. Subplot (d): Why were no scenarios excluded (colored white) due to biomass being too low for grazing?

27) L330: "significantly" implies a statistical test that I don't think was performed.

28) L336: Replace "primarily" with "entirely".

29) (L338) Fig. 9b, d: Why were no scenarios excluded (colored white) due to biomass being too low for grazing?

30) L343-344: This statement doesn't seem to be true for grazing alone, except maybe for the Caatinga site.

31) (L346) Fig. 10:

a. This figure was very confusing at first. I was eventually able to understand it: The colors only represent N *loss* mechanisms, so the top of the bar is N *inputs* (as the authors mention), and then the colors go down from there, with the bottom of the bar representing the N *balance*. I think that last point should be explained in the caption. It is much more typical for these kinds of plots to have inputs stacking on top of the zero line and losses below, with a star or something to note the net flux.

b. And indeed, that's what you do in Fig. 11! I strongly suggest switching Fig. 10 to this format, using the same colors for deposition and BNF as in Fig. 11

c. Dark blue color label should be "Nitrification + denitrification".

32) L351-352: Are you saying that both BNF and litterfall are directly attributable to the plant C and N pools? Is that because LPJmL doesn't represent asymbiotic BNF?

33) L354: "which in turn are affected by the same fate." What is the antecedent of "which" here? "nitrogen uptake"? In that case, "are" should be "is".

34) L355-356: My interpretation is that this is because low-biomass plants can't "pay" for much symbiotic BNF—is that right? This should be explained.

35) L356:
a. This statement doesn't make sense in the context of Fig. 11, which deals with soil N only, not ecosystem N.

b. The "primarily due to minimal grazing" part of this statement is not supported by Figs. 10 or 11. Fig. 10c suggests that most N losses from the Caatinga system are due to leaching, not grazing (harvest N loss). While the harvest N bars are needed in most cases to drop the net flux below the zero line in Fig. 10c, that figure *only* shows the result for the Caatinga with grazing on. It can't be safely assumed that ecosystem or soil N flux would be positive without grazing, because without grazing many parts of the system are changed. The authors performed more experiments than are shown here, but if those experiments support this assertion, they should be presented somewhere and cited here.

36) (L358) Fig. 11:
a. Please add a star or something to each bar representing the net N flux.

b. Dark blue color label should be "Nitrification + denitrification".

37) L369-371: What drives the ratio of allocated C:N and how it changes after disturbance?

38) L371: Again, what "deficit"? Are plants N-limited?

39) L378-382:
a. Note that it's still not a beneficial effect even in the Pampas. I think "in wetter regions like the Pampas, fire and grazing can coexist with higher vegetation

productivity" should thus be struck, or at least strongly modified. Even the highest-biomass cell in Figs. 5c-d (i.e., grazing + fire in the Pampas) represents a 17% reduction of leaf biomass relative to the control treatment.

    b. "favourably" should be "favourable".

40) L393: "Sec. A" should be "Appendix A". Also, why is Fig. A1 in an Appendix Section all by itself? Wouldn't it be simpler to just have one Appendix with all additional results? In addition, am I right that this is the first time Fig. A1 is mentioned anywhere? In that case it should come later in the Appendix, because other Appendix figures have already been mentioned.

41) L409: "an initial reduction in intake" when? With the introduction of fire?

42) L413: "this example"—which?

43) L430: "relevant" is probably not the right word, since in the next sentence you say that's not how it works in the real world.

44) L435:

    a. Surely not "all plant biomass is treated as fuel"—maybe just aboveground?

    b. "moribund" is probably not the right word. At least, it's not very clear how "nearly dead" plant parts are most affected.

45) L438: "functionality specially" should be "functionality, especially".

46) L448: "parsimonious" is probably not the right word.

47) Throughout:

    a. Use of "significant" implies a statistical test when none was performed. This mostly happens in figure captions but is also present in the main text.

---

## Referee Comment (RC2)

This manuscript presents a timely and relevant study that integrates a pasture-burning date algorithm into the LPJmL5 dynamic global vegetation model to assess the coupled impacts of fire and grazing on ecosystem carbon and nitrogen cycling in Brazilian grasslands. Using Chalumeau algorithm for climate-driven, management-based fire timing represents a valuable methodological advancement, particularly in regions where human land use plays a dominant role in shaping fire regimes. The paper is well-motivated, generally well-written that emphasize the interplay between grazing pressure, burning frequency, and biogeochemical feedback.

However, the study also has important limitations that should be more clearly acknowledged. Chief among these is the absence of any experimental or observational data used for model calibration or validation, despite the strong claims made about nitrogen and C: N dynamics. Additionally, several key biogeochemical processes—such as nitrogen recycling via manure and biological nitrogen fixation—are either under-described or not quantitatively supported. Such issues do not undermine the value of the study as a modelling exploration. However, they must be addressed more explicitly. Please see my detailed comments below for specific suggestions on how to strengthen the manuscript.

1.) Provide a supplementary table that lists every fixed parameter, its value, units, and reference/justification. This will satisfy transparency and help future users turn the feature on in standard LPJmL runs.

2.) In the nitrogen-budget discussion the authors state that nitrogen inputs "consist of BNF and atmospheric deposition," and the decline in soil N under burning / grazing is "driven primarily by reductions in BNF. However, there is no description of how BNF is computed (symbiotic vs. asymbiotic, dependence on plant N demand, moisture, temperature, etc.) nor any parameter values.

   A) Include the key BNF parameters in the requested parameter table (fixation efficiency constants, maximum rates, etc.).
   B) If possible, show one sensitivity test (e.g. ±20 % maximum BNF rate) to demonstrate that the qualitative C: N conclusions are robust.

3.) Nitrogen cycling clarity - The current text never states what fraction of grazed N is returned as dung/urine or how urine is handled, calling the soil-N results into question.
   A. Give the exact fractions and pathways in Sect. 2.1.2.
   B.  Cite a data source and show that the chosen value falls within the empirical 70–95 % range.
   C.  Run a quick sensitivity test (e.g. +20 % manure-N return) and state whether conclusions change.

4.) This study does not use any experimental field data for calibration or validation. Its findings are based entirely on mechanistic simulations, and while it emphasizes the importance of future field research, it would benefit from moving that discussion into a dedicated 'Limitations and Outlook' subsection

---

## Author Comment (AC1)

We thank the reviewer for their thorough and constructive feedback. We are pleased that the methodological contribution of this study and the relevance of integrating fire and grazing management into LPJmL were appreciated. Below, we provide a point-by-point outline of how we plan to address each of the reviewer's comments in the revised manuscript.

All comments related to the figures are grouped together at the end. Minor comments, such as typos or small wording changes, are not listed individually but will be integrated as proposed.

**Major comments:**

> **The main thing I'm concerned about here is how grazed nitrogen is handled in the model. There's no citation in the Methods, as far as I noticed, pointing to information about how LPJmL grazing works at all, actually. N isn't mentioned alongside C at L131-133 as being partially returned to the soil via animal waste after grazing. That initially led me to think it wasn't. Then at L267, "manure (in grazing systems)" is mentioned as one input to the soil. But what fraction of the consumed N? And is it only feces or is it also urine?**
>
> **[…]**
>
> **If a realistic fraction of the grazed N isn't returned to the soil, I have serious doubts about the LPJmL model's fitness for the purpose of analyzing impacts of grazing on soil N and thus leaf C:N ratio too. See Selbie et al. (2015, DOI 10.1016/bs.agron.2014.09.004): "Ruminants excrete as much as 70–95% of the nitrogen (N) they consume."**

Thank you for highlighting this important point. This issue was also raised by the Reviewer 2, and we have provided a detailed response addressing it in our reply. For completeness and clarity, we kindly refer Reviewer 1 to that response, where we clarify that 66.7% of the nitrogen contained in the grazed biomass is returned to the soil as manure including feces and urine.

> **Then again, in Fig. 10, neither manure nor urine are explicitly mentioned. They might be included in "harvest N" with that color showing net harvest N loss, but that's not explained. And finally, Fig. 11 does have manure again.**

Thank you for pointing this out. Figure 10 represents the ecosystem nitrogen budget, which includes only external nitrogen fluxes. Since manure is not treated as an external nitrogen input in our model setup, but a conversion of plant nitrogen it is not included in this figure. The component 'harvest N' in Fig. 10 contains only the nitrogen that is grazed and removed from our model domain so that it does not include the nitrogen in the manure.

In contrast, Figure 11 focuses on the soil nitrogen budget, which explicitly accounts for manure as an input to the soil nitrogen pool. The distinction between these two perspectives is clarified in the figure captions, both of which refer to the section "Nitrogen in- and out-fluxes" (previously Section 2.4.3, now moved to Section 2.1.4), where the composition of the budgets is described in detail.

To clarify the point raised regarding manure, we have added explicitely that manure is excluded from the ecosystem budget in its description :

*The ecosystem nitrogen budget is determined by the balance of nitrogen input, biological nitrogen fixation (BNF) and atmospheric deposition, and nitrogen outputs, including leaching,*

*denitrification, volatilisation, plant uptake, harvest nitrogen (in grazing systems excluding the part returned to the soil as manure) and NOX emissions from fire (in burning scenarios).*

**If the N analyses are kept, some text needs to be added to the Methods describing biological nitrogen fixation (BNF) in LPJmL. Are both symbiotic and asymbiotic BNF represented? How do they work?**

We thank the reviewer for raising this important point. This issue was also brought up by Reviewer 2, and we have provided a detailed explanation in our response to their comment. In short, we clarify how BNF is computed in LPJmL providing the equation and how we will explained it in the manuscript.

**Other comments**

**1) I would like to see some text added to the Discussion or Conclusion about what work would be needed in order for this feature to become commonly enabled in LPJmL runs. Is it just livestock density maps (both historical and for future scenarios) that are holding it back?**

The main prerequisite for applying the pasture burning version of LPJmL and the Chalumeau module in other regions is a solid understanding of local burning practices. In this study, we focused on Brazil, where we could base the implementation on existing literature and documented fire management practices. While the pasture burning functionality itself is flexible and can be technically applied to other regions, its meaningful use requires region-specific calibration regarding fire season. Indeed, the broader application of the Chalumeau algorithm would require a careful reassessment, especially in temperate and polar regions. The algorithm relies heavily on climate inputs and seasonal patterns, which may not translate well across different biomes without further development and validation. We have added a few lines addressing these aspects in the newly titled "Limitations and Outlook" section of the manuscript (l. 441):

*Applying the pasture burning version of LPJmL and Chalumeau module beyond Brazil would require region-specific information on fire management practices. While the current implementation is technically flexible, meaningful application in other regions depends on adapting fire-use assumptions. In particular, the Chalumeau algorithm, which is driven by seasonal and climatic constraints, should be carefully re-evaluated before application to temperate or polar regions.*

**2) The Appendices are strange. Appendix A has only one figure—why not combine the two Appendices? That figure, Fig. A1, is also the very last to be mentioned in the main text (and only as "Sec. A", not "Fig. A1"); it should thus be last in the Appendix as well. Finally, there are four Appendix figures not mentioned in the main text: Figs. B2–3 and B5–6.**

We acknowledge the inconsistency in appendix structure and figure referencing. Appendices A and B have been merged into a single section. We have revised the order of figures so that Fig. A1 (currently referenced last) appears at the end and we have ensured that all appendix figures are explicitly referenced in the main text with appropriate labels.

**6) L157-166 (Sect. 2.2.1): Since the Brunel et al. (2021) and Waha et al. (2012) papers are not open-access, more detail should be given here (or in an Appendix/Supplement) on the Chalumeau algorithm and its implementation. For instance, how often is the burning date updated? Does it use a rolling window to calculate seasonality variables?**

**What's the difference between the "burning date" vs. the strategies (e.g. "early spring")? Etc.**

Thank you for the suggestion. We will include additional details on the Chalumeau module in the Appendix to improve clarity and accessibility. Briefly, Chalumeau identifies the dormant season (DS) and extracts burning dates using daily temperature and precipitation data. A 10-day moving average for temperature-driven seasonality and a 10-day moving cumulative sum for precipitation-driven seasonality are used to identify the dormant season as either the winter or dry period. Then, depending on the chosen burning strategy, a single burning date is extracted per grid cell and DS.

**8) L224-233: Is other fire allowed to happen during these experiments? I.e., are the only ignitions allowed due to intentional pasture management burning, or is the rest of SPITFIRE operating at the same time?**

Only the intentional burnings are applied to the pasture during the simulated experiment. Natural fires do not occur. We clarify this point in the Section 2.3.2, Model configuration and experimental setup adding this sentence (l.229) :

*Burning practices are the only fires applied to the pasture during the experiment.*

**9) L227-229: It's not until this sentence until I understood what this paragraph was supposed to be describing; until then I was pretty confused. Please move it to the top of the paragraph and edit as needed for flow.**

Upon re-reading the section, we agree that the structure was somewhat confusing. Following your suggestion, we have moved the respective sentence to the beginning of the section (l.209), hoping that this will better guide the reader through the description of the experimental setup.

**10) L230-231: I'm not sure I understand this correctly. How many replicates does this result in? 2 + 5 + 10 = 17?**

Thank you for pointing this out. Indeed, the number of replicates corresponds to the length of the burning frequency: the 2-year frequency scenario includes 2 replicates (starting in year 0 and year 1), the 5-year frequency includes 5 replicates (starting from year 0 to year 4), and the 10-year frequency includes 10 replicates (starting from year 0 to year 9). We have clarified this in the manuscript by adding the following sentence in (l.231):

*This results in 2, 5, and 10 replicates for the 2, 5, and 10 year burning frequency scenarios respectively.*

**11) L233: Are the four strategies something that the Chalumeau algorithm produces for each gridcell? Or are they things you switch between for different experiments?**

Thank you for this question. The burning strategy is provided as an input parameter to the Chalumeau algorithm for each experiment. Based on the selected strategy, Chalumeau determines the corresponding burning dates. We will clarify this point in the Supplement, as mentioned in our response to comment 6.

**14) L262-268 (Sect. 2.4.3): Why is this in the "Post-processing" section? It would be more appropriate near Sect. 2.1.3 ("Soil nitrogen pools").**

We agree that the paragraph does not fit well within the "Post-processing" section. However, since it addresses the broader ecosystem nitrogen budget rather than soil nitrogen pools alone, we have

moved it after Section 2.1.3. We think that this new placement ensures better alignment with the content and improves the overall structure of the Methods section.

**17) L288: What nitrogen deficit? It hasn't been previously established that NPP is Nlimited under any conditions or treatment.**

Thank you for this observation. We have revised the sentence to clarify that the nitrogen deficit in leaves is a result of the burning practices (l.288):

*Burning practices lead to a nitrogen deficit in leaves, significantly affecting the nutrient balance of the vegetation.*

**19) L297–299: How is this result possible?**

Thank you for your comment. This result is indeed discussed in more detail in Lines 406–419 of the Discussion section. To summarize it, the difference in responses between biomass pools (AGB) and fluxes (dry matter intake) arises from their distinct sensitivities to disturbances. While AGB shows a gradual, cumulative decline over time due to repeated grazing and burning, intake (a flux) initially drops but stabilizes as the system adapts. This is because intake responds more directly to the balance between net growth and losses: as biomass decreases, fire intensity and thus disturbance also diminishes, allowing intake to reach a new, stable equilibrium despite lower overall biomass.

To improve clarity for the reader, we now include a brief indication in the Results section to highlight the key factor behind this result and guide the reader to the relevant discussion (l.299).

*This counter-intuitive pattern results from differences in the response of biomass pools and fluxes to repeated disturbances, as discussed in Section 4.*

**20) (L298) Fig. 4d: Why were no scenarios excluded (colored white) due to biomass being too low for grazing?**

As mentioned in Section 2.4.1, scenarios are excluded when the average dry matter intake falls below 80% of the livestock's feed requirement. Biomass levels can be lower, but still sufficient to meet this threshold. The reasoning behind this result is discussed in Section 4 (lines 406–419) and summarized in our response to Comment 19.

**21) L304:**
**a. Why does recently-established grazing have such a higher average dry matter intake?**
**b. Are these numbers for the "no burning" treatment specifically?**

Thank you for your question. Yes, these values refer specifically to the no burning treatment. The higher average dry matter intake under recently-established grazing results from the fact that vegetation biomass is still relatively high at the beginning of the simulation, before the full effects of grazing disturbance accumulate. We have clarified this in the manuscript (l.304):

*Under no burning conditions, the average dry matter intake is 309 g m2 for recently-established grazing and 42 g m2 for pre-established grazing. The higher intake in the recently-established scenario reflects the initially high vegetation biomass before long-term grazing effects reduce plant availability.*

**23) L316-317: What is this "optimum" value? Optimum for what?**

Thank you for the comment. The "optimum" value refers to the soil C:N ratio considered ideal for promoting healthy nutrient cycling . We have clarified this in the revised text (l.315):

*In fact, even in undisturbed scenarios, the soil in the Cerrado site is not rich in nitrogen, as indicated by a soil C:N ratio of 16.65, which is above the target value of 15 considered an optimum for maintaining nitrogen availability to plants.*

**25) L321: What "initial nitrogen deficit"? Deficit relative to what?**

Thank you for the comment. We have clarified in the manuscript that we are referring to a soil nitrogen deficit (l.321):

*However, in the recent disturbance scenario, we notice that the introduction of burning practices helps to alleviate the initial soil nitrogen deficit, decreasing the C:N ratio up to 1.6% without grazing and with frequent burning.*

**32) L351-352: Are you saying that both BNF and litterfall are directly attributable to the plant C and N pools? Is that because LPJmL doesn't represent asymbiotic BNF?**

Thank you for pointing this out. The original sentence was indeed misleading. We intended to say that BNF and litterfall originate from the vegetation pools. In the model, both are attributed to the soil nitrogen budget: BNF is allocated to the $NH_4^+$ pool, while litterfall contributes to the SOM pool. We have revised the sentence accordingly (L351–352):

*In the case of the soil nitrogen budget, only BNF and litterfall are directly derived from the plant carbon and nitrogen pools and are the primary input fluxes of nitrogen.*

For clarification regarding BNF, LPJmL produces a crude estimate of total BNF without distinguishing between fixation mechanisms and without accounting for the energetic cost to the plant for nitrogen fixation.

**35) L356:**
**a. This statement doesn't make sense in the context of Fig. 11, which deals with soil N only, not ecosystem N.**
**b. The "primarily due to minimal grazing" part of this statement is not supported by Figs. 10 or 11. Fig. 10c suggests that most N losses from the Caatinga system are due to leaching, not grazing (harvest N loss). While the harvest N bars are needed in most cases to drop the net flux below the zero line in Fig. 10c, that figure only shows the result for the Caatinga with grazing on. It can't be safely assumed that ecosystem or soil N flux would be positive without grazing, because without grazing many parts of the system are changed. The authors performed more experiments than are shown here, but if those experiments support this assertion, they should be presented somewhere and cited here.**

Thank you for highlighting this inconsistency. We agree that the original statement was misleading in the context of Fig. 11. As shown in Fig. 10c, leaching is indeed the main driver of net nitrogen loss in the Caatinga system, not grazing. Furthermore, simulations without grazing also show a net nitrogen loss, indicating that grazing is not the primary cause of the negative nitrogen balance. While these additional results support our statement, we have chosen not to include them in the manuscript to maintain focus and keep the manuscript concise. However, we have revised the text to better reflect the findings shown in Fig. 10c and avoid unsupported claims (l.356) :

*Even without fire practices, the ecosystem experiences a net nitrogen loss primarily due to the leaching (Fig.10 c).*

**37) L369-371: What drives the ratio of allocated C:N and how it changes after disturbance?**

The allocation process follows a strict, non-linear scheme influenced by changes in water and nitrogen availability. We prescribe a target range for the C:N ratio in leaves and roots. Variations in water and nitrogen affect the actual allocation to ensure values remain within this prescribed C:N ratio range. At the same time, water and nitrogen availability influence leaf mass, thereby altering the leaf-to-root carbon mass ratio. Disturbances therefore do not primarily target the C:N ratio itself but rather the leaf-to-root ratio, which in turn can affect the C:N ratio. We will clarify this mechanism in the revised manuscript to make it more explicit.

**47) Throughout:**
**a. Use of "significant" implies a statistical test when none was performed. This mostly happens in figure captions but is also present in the main text**

Thank you for pointing this out. We have revised the text and figure captions to avoid the use of "significant" in contexts where no statistical test was performed, in order to prevent any confusion.

**Comments related to figures**
Redundant comments have not been reported to avoid repetition.

**3) All figures except Fig. 1, or at least certain labels in those figures, seem have JPEG artifacts. In most cases these should be replaced with entirely vector-based figures (.eps or .pdf). Failing that, PNG should be used. JPEG should only be used for photos (and don't just convert JPEGs to PNG!).**

We will revise all figures accordingly. Specifically we will replace current figures with vector formats (.EPS or .PDF) wherever possible.

**4) The "matrix"-type figures (Figs. 4–9) need a fair amount of work:**

**a. All of these figures should have color bars. It should also be made obvious when subplots share a color bar.**

**b. There are many cells in these figures with black text on a dark background. This should be avoided, for instance by adding a white outline or "glow" around all text that overlays a non-white background.**

**c. Some of these seem to be true values, while others are changes relative to a baseline. This is hard to keep track of and introduces an extra mental load in interpreting them. Please consider standardizing on one or the other.**

**d. In some of these, white represents "excluded because of insufficient biomass for grazing," whereas in others it's burgundy. This should be standardized to burgundy (or even better for colorblind readers, black). I say this because white is confusing: In Figs. 4-5, the lightest color (yellow) is low impact, the darkest color (dark red) is high-impact, and pure white is the highest impact of all. The color scale goes light-dark-light.**

**e. Most of the cells in these matrices represent a "bad" impact, so it makes sense they are represented by a yellow-red color scale. However, some represent a "good" impact —e.g., some treatments in Fig. 8 showing soil N enrichment. In such cases, they should not be on the same color scale, because they are qualitatively different. Something other than yellow-red should thus be used—e.g., blues.**

**f. An extra column should be added to the right side of every matrix giving the results with no fire.**

We appreciate the detailed suggestions and will incorporate them into the revised version. We are aware that the complexity of these figures, due to the large volume of data, can hinder readability. Improving their clarity is a priority, as it will enhance both the accessibility and the overall understanding of the paper.

We fully agree that the current mix of true values and relative changes can be difficult to follow and adds unnecessary cognitive load. Following your suggestion, we have standardized all results to be expressed as relative values, using the no burning scenario as the reference. Since the focus of the figure is on the impact of burning, we believe that expressing results as percentage changes relative to the reference scenario improves clarity and facilitates interpretation.

**22) (L311) Fig. 6:**
**a. It'd be nice to have the site names in the figure title, as is done for similar figures.**

Thank you for pointing this out. Initially, the site names were placed to the side of the figure, since Fig. 6 exceptionally displays results for two sites simultaneously. However, to improve consistency and clarity, we have now also included the site names in the figure title, in line with the formatting of similar figures.

**b. C:N normalized to a percentage feels wrong, I think because it's a result of both the numerator and the denominator. Consider changing to actual values instead of percentage.**

We understand the concern. Our choice to normalize C:N ratios was motivated by the need to focus on the relative impact of grazing and burning across sites with very different baseline conditions. As explained in Section 2.4.2, the C:N ratios are normalized using the reference C:N value for each site under undisturbed conditions. This approach enables meaningful comparisons of treatment effects across heterogeneous environmental contexts.

That said, we will clarify this reasoning more explicitly in the caption and/or the main text to avoid confusion.

**31) (L346) Fig. 10:**
**a. This figure was very confusing at first. I was eventually able to understand it: The colors only represent N loss mechanisms, so the top of the bar is N inputs (as the authors mention), and then the colors go down from there, with the bottom of the bar representing the N balance. I think that last point should be explained in the caption. It is much more typical for these kinds of plots to have inputs stacking on top of the zero line and losses below, with a star or something to note the net flux.**
**b. And indeed, that's what you do in Fig. 11! I strongly suggest switching Fig. 10 to this format, using the same colors for deposition and BNF as in Fig. 11**
**c. Dark blue color label should be "Nitrification + denitrification".**

Thank you for this valuable feedback. We agree that the current version of the figure can be confusing at first glance. As mentioned in our response to Reviewer 2, we are planning to revise the figure to adopt a more conventional layout.

---

## Author Comment (AC2)

We thank the reviewer for their thoughtful and constructive comments, which help us improve the clarity, transparency, and scientific robustness of the manuscript. We appreciate the recognition of the methodological contribution and relevance of our study. Below we outline how we plan to address each of the specific comments in the revised version of the manuscript.

>  ***1.) Provide a supplementary table that lists every fixed parameter, its value, units, and reference/justification. This will satisfy transparency and help future users turn the feature on in standard LPJmL runs.***

We fully agree with the importance of parameter transparency. Given the large number of parameters involved in the LPJmL model, we would follow this approach:

The general model parameters, i.e. unmodified LPJmL5 settings, are already available in the publicly accessible LPJmL-Pasture-Burning version repository. A direct link to the code is provided at the end of the manuscript in the section code availability. In addition, we have added a reference to it in the Model configuration and experimental setup subsection as follow (l. 234):

*A complete list of the general parameters used during the simulations can be found in the configuration files of the model, which are available via the link provided in the Code Availability section.*

To complement this, we have added a new supplementary table listing all parameters that are specific and most relevant to this study. This includes parameters related to the Chalumeau algorithm, the pasture and grazing management system (e.g., fraction of manure returned to the soil), and the nitrogen cycle (including BNF). The table reports parameter names, values, units, and where available the corresponding references or justifications. This table is included in the appendix as Table S1.

We believe this approach ensures both completeness and accessibility while keeping the main manuscript concise.

>  ***2.) In the nitrogen-budget discussion the authors state that nitrogen inputs "consist of BNF and atmospheric deposition," and the decline in soil N under burning / grazing is "driven primarily by reductions in BNF. However, there is no description of how BNF is computed (symbiotic vs. asymbiotic, dependence on plant N demand, moisture, temperature, etc.) nor any parameter values.***

We thank the reviewer for pointing out the lack of detail regarding the representation of BNF in the manuscript. We acknowledge that the current version does not provide sufficient information on this important process.

The current implementation in our model is relatively simple. BNF is calculated from the 20-year average of annual evapotranspiration following the function from Cleveland et al. (1999). The resulting BNF is added to the $NH4+$ pool in the first soil layer. The description of this process as well as the equation are added in the section 2.1.3, Soil nitrogen pools, as follow (l. 140):

*BNF is calculated from the 20-year average of annual evapotranspiration following the function from Cleveland et al. (1999). The resulting BNF is added to the $NH+4$ pool in the first soil layer.*

$$BNF = \begin{cases} \max(0, (0.0234 \cdot etp - 0.172)/10/365) & \text{if } C_{root} > 20\,\mathrm{g\,C\,m}^{-2} \\ 0 & \text{otherwise.} \end{cases}$$

We have also noticed that Figure 10 might be difficult to interpret, particularly regarding the nitrogen inputs, as they are not explicitly shown. The original figure focuses solely on the nitrogen deficit. To improve transparency and enhance understanding, we have decided to revise the figure to display the nitrogen inputs, namely BNF and atmospheric deposition. The revised figure follows the same structure as Figure 11, with nitrogen inputs shown above the zero axis and nitrogen outputs below.

> **A) Include the key BNF parameters in the requested parameter table (fixation efficiency constants, maximum rates, etc.).**

As mentionned in the answer of the point 1, all parameters regarding BNF are included in the requested parameter table.

> **B) If possible, show one sensitivity test (e.g. ±20 % maximum BNF rate) to demonstrate that the qualitative C: N conclusions are robust.**

We appreciate the reviewer's suggestion to include a sensitivity test to assess the robustness of our conclusions. However, running additional simulations is computationally expensive, especially given the scale and complexity of our current model setup.

That said, we can reasonably anticipate the direction of the effect: increasing the maximum BNF rate would enhance BNF fluxes, thereby increasing nitrogen inputs into the $NH_4^+$ soil pool. However, this does not translate linearly into plant growth, as higher nitrogen availability also accelerates soil transformation processes such as immobilization, denitrification, and leaching.

Conversely, a decrease in the maximum BNF rate would reduce nitrogen inputs, potentially accelerating the decline of soil nitrogen pools.

While we are confident that such a test would not change the qualitative conclusions of our study, we acknowledge its potential value and consider it a relevant avenue for future work.

> **3.) Nitrogen cycling clarity - The current text never states what fraction of grazed N is returned as dung/urine or how urine is handled, calling the soil-N results into question.**
>
> **A. Give the exact fractions and pathways in Sect. 2.1.2.**
>
> **B. Cite a data source and show that the chosen value falls within the empirical 70–95 % range.**

We thank the reviewer for pointing out this omission and fully acknowledge the lack of detail in the current manuscript. This was indeed an oversight on our part.

In our model, 66.7% of the nitrogen contained in grazed biomass is returned to the soil via excreta, primarily as urine. This value lies within the lower range of empirically observed values (typically 70–95%), which reflects the fact that livestock are not permanently present in the field. Our

assumption accounts for periods when animals are housed or moved, during which excreta are not deposited directly onto the grazed area.

We have included this information in the Section 2.1.2, Managed grassland and grazing, as follow (l. 134) :

*For nitrogen, 66.7% of the grazed nitrogen is returned to the soil, primarily as urine and dung, and is allocated to the NO3 pool in the first soil layer. This value lies at the lower end of the empirically observed range of 70–95% reported by Selbie et al. (2015), reflecting the fact that cattle are not continuously present on the pasture. Periods during which livestock are housed or moved off-site are thus taken into account by this assumption.*

**C. Run a quick sensitivity test (e.g. +20 % manure-N return) and state whether conclusions change.**

We thank the reviewer for this valuable suggestion. As previously mentioned in our response regarding the BNF sensitivity test, running additional simulations is computationally expensive.

Nonetheless, we can anticipate the likely effects of a ±20% change in the manure-N return fraction. Increasing the proportion of nitrogen returned to the soil would enhance nitrogen inputs to the $NO_3^-$ soil pool. Conversely, a reduction in this return fraction would decrease nitrogen inputs, possibly accelerating the decline of soil nitrogen pools. However, as already explained for the BNF, this does not translate linearly into plant growth, as higher nitrogen availability also accelerates soil transformation processes such as immobilization, denitrification, and leaching.

**4.) This study does not use any experimental field data for calibration or validation. Its findings are based entirely on mechanistic simulations, and while it emphasizes the importance of future field research, it would benefit from moving that discussion into a dedicated 'Limitations and Outlook' subsection**

Thank you for this helpful suggestion. We have renamed the "Conclusion" section to "Limitations and Outlook" and merged it with the final part of the Discussion (starting from l. 427). In this revised section, we have added a paragraph explicitly addressing the lack of experimental field data for calibration or validation of the burning practices. While acknowledging this limitation, we have also mentionned that the LPJmL model and its internal dynamics have been previously validated, and we have provided the corresponding reference (l. 442):

*This study does not rely on experimental field data to calibrate or validate the results specific to burning practices. Indeed datasets documenting the impact of burning on vegetation structure, yields, and soil carbon and nitrogen content are either unavailable or entirely lacking for Brazil and the various locations analysed here. However, the LPJmL model itself, along with the underlying process dynamics it simulates, has been previously validated (Schaphoff et al., 2018a, b; von Bloh et al.,2018a, b).*